# Neural Task Synthesis for Visual Programming

**Victor-Alexandru Pădurean**                                    *vpadurea@mpi-sws.org*
*Max Planck Institute for Software Systems*

**Georgios Tzannetos**                                            *gtzannet@mpi-sws.org*
*Max Planck Institute for Software Systems*

**Adish Singla**                                                 *adishs@mpi-sws.org*
*Max Planck Institute for Software Systems*

**Reviewed on OpenReview:** *https://openreview.net/forum?id=aYkYajcJDN*

## Abstract

Generative neural models hold great promise in enhancing programming education by synthesizing new content. We seek to design neural models that can automatically generate programming tasks for a given specification in the context of visual programming domains. Despite the recent successes of large generative models like GPT-4, our initial results show that these models are ineffective in synthesizing visual programming tasks and struggle with logical and spatial reasoning. We propose a novel neuro-symbolic technique, NEURTASKSYN, that can synthesize programming tasks for a specification given in the form of desired programming concepts exercised by its solution code and constraints on the visual task. NEURTASKSYN has two components: the first component is trained via imitation learning procedure to generate possible solution codes, and the second component is trained via reinforcement learning procedure to guide an underlying symbolic execution engine that generates visual tasks for these codes. We demonstrate the effectiveness of NEURTASKSYN through an extensive empirical evaluation and a qualitative study on reference tasks taken from the *Hour of Code: Classic Maze* challenge by Code.org and the *Intro to Programming with Karel* course by CodeHS.com.

## 1 Introduction

Recent advances in generative AI have demonstrated impressive performance in a variety of domains, including visual art and music creation (Dong et al., 2018; Briot et al., 2020; Suh et al., 2021; Ramesh et al., 2021; Rombach et al., 2022), medicinal chemistry synthesis (Schneider et al., 2020; Walters & Murcko, 2020; Tong et al., 2021; Gao & Coley, 2020), and AI-enhanced programming (Finnie-Ansley et al., 2022; Leinonen et al., 2023; Chen et al., 2021; Feng et al., 2020; Phung et al., 2023a). These successes are, in part, driven by advanced capabilities of deep generative models, such as Stable Diffusion (Rombach et al., 2022), ChatGPT (OpenAI, 2023a), and GPT-4 (OpenAI, 2023b). These advancements also hold great promise in enhancing education, for instance, by generating personalized content and new practice tasks for students allowing them to master required concepts (Sarsa et al., 2022; Tate et al., 2023; Baidoo-Anu & Owusu Ansah, 2023; Lim et al., 2023; Phung et al., 2023b).

In this paper, we explore the role of generative AI in visual programming domains used for introductory programming. Popular domains, such as Scratch (Resnick et al., 2009), *Hour of Code:Maze Challenge* by Code.org (HoCMaze) (Code.org, 2013b;a), and Karel (Pattis et al., 1995), have become an integral part of introductory computer science education and are used by millions of students (Code.org, 2013a; Wu et al., 2019; Price & Barnes, 2017). In existing visual programming platforms, programming tasks are hand-curated by tutors and the available set of tasks is typically very limited, posing a major hurdle for novices in mastering the missing concepts (Zhi et al., 2019; Ahmed et al., 2020). To this end, we seek to design generative models that can automatically synthesize visual programming tasks for a given specification (see Figure 1).

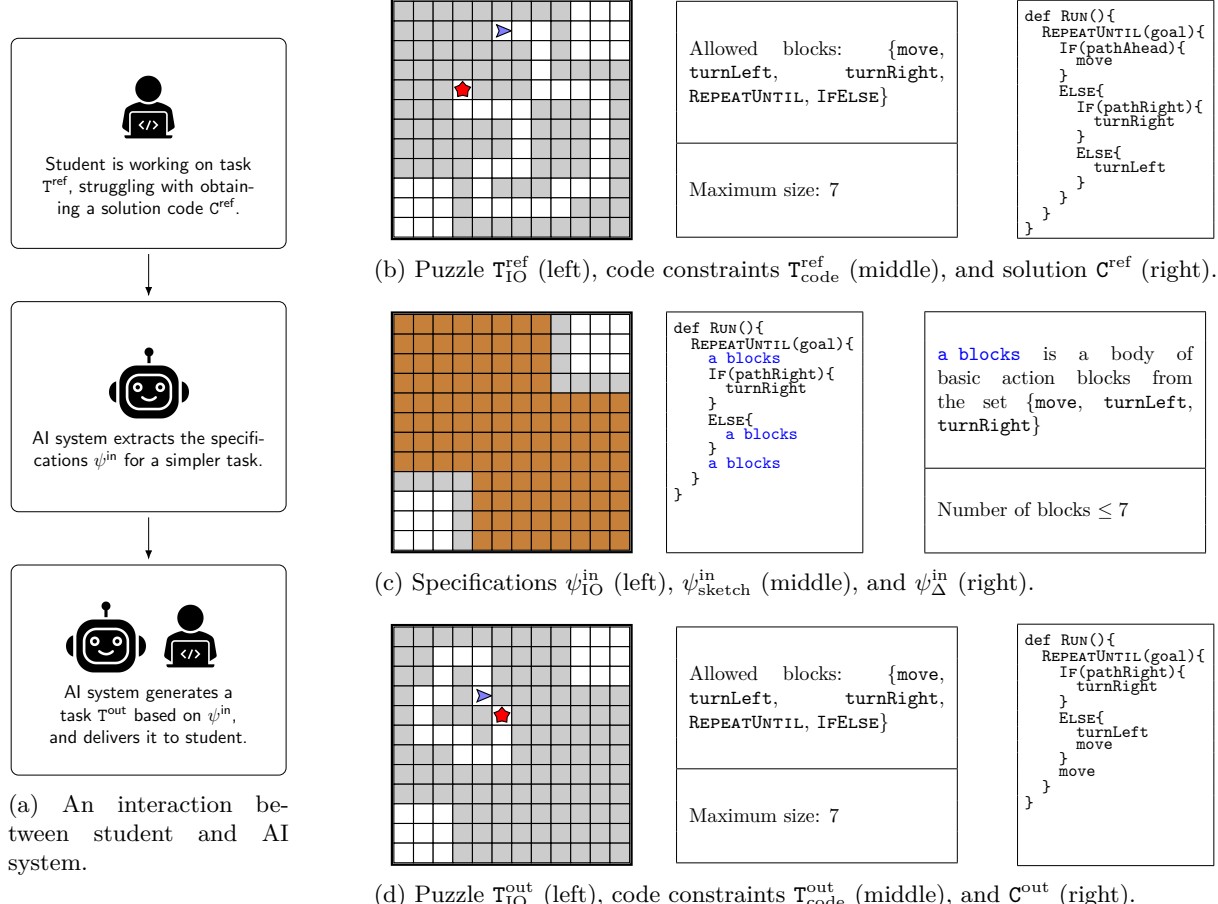

(a) An interaction between student and AI system.

(b) Puzzle $\text{T}^{\text{ref}}_{\text{IO}}$ (left), code constraints $\text{T}^{\text{ref}}_{\text{code}}$ (middle), and solution $\text{C}^{\text{ref}}$ (right).

(c) Specifications $\psi^{\text{in}}_{\text{IO}}$ (left), $\psi^{\text{in}}_{\text{sketch}}$ (middle), and $\psi^{\text{in}}_{\Delta}$ (right).

(d) Puzzle $\text{T}^{\text{out}}_{\text{IO}}$ (left), code constraints $\text{T}^{\text{out}}_{\text{code}}$ (middle), and $\text{C}^{\text{out}}$ (right).

Figure 1: **(a)** Illustration of an AI system helping a student, inspired by (Ghosh et al., 2022). The AI system supports the student when trying to solve $\text{T}^{\text{ref}}$. The system first extracts a specification for a simpler task that can be provided to help the student. It then generates a new task based on that specification and delivers it to the student. **(b)** shows reference task $\text{T}^{\text{ref}}$ along with solution code $\text{C}^{\text{ref}}$. **(c)** shows specifications $\psi^{\text{in}}$ extracted by AI system. **(d)** shows task $\text{T}^{\text{out}}$ and code $\text{C}^{\text{out}}$ synthesized by AI system.

## 1.1 Motivation and Overview

Our work is motivated by existing literature on AI-driven educational systems that seek to provide feedback to students who are struggling while solving a programming task. For instance, (Zhi et al., 2019) studies feedback provided in the form of worked examples as demonstration and (Ghosh et al., 2022) studies feedback provided in the form of new simpler tasks for adaptive scaffolding. We have illustrated the interaction between such a system and a student in Figure 1. Previous works have focused on studying how to adapt the feedback type according to the student's needs. We complement these works by focusing on generating on-the-fly the desired examples and simpler tasks that the system could provide to the student. In our work, we consider that a desired feedback format for a student is an input specification in the form of puzzle layout, code structure, code size, and selected programming concepts. Since the search space defined by these input specifications is potentially unbounded, designing AI systems that can synthesize correct tasks in real-time for any input specification is critical to cater to a diverse set of students' behaviors.

As a natural approach, one might be tempted to employ state-of-the-art large language models (LLMs) to generate a visual programming task by providing task synthesis specification as a prompt. In particular, models like GPT-4 are trained on multi-modal data, including text, code, and visual data, and hence it seems a suitable technique for reasoning about visual programming tasks (OpenAI, 2023b; Bubeck et al., 2023). However, our results show that these models are ineffective in synthesizing visual programming tasks and struggle with logical and spatial reasoning, as also indicated in recent literature on state-of-the-art

models (Bang et al., 2023; Bubeck et al., 2023; Valmeekam et al., 2022; Huang & Chang, 2022); see Section 5 for detailed discussion. In general, a major challenge in using purely neural generative models for synthesizing visual programming tasks is that the generative process is highly brittle – even a small modification in the output task could make it invalid or semantically incorrect w.r.t. the input specification (Ahmed et al., 2020).

As an alternative to neural generative models, we could rely on symbolic generative methods driven by search and planning algorithms to generate content that matches a specification. Several works have shown the efficacy of symbolic methods to generate new tasks in various educational domains, e.g., algebra exercises (Singh et al., 2012; Gulwani, 2014), geometric proof problems (Alvin et al., 2014), natural deduction (Ahmed et al., 2013), mathematical word problems (Polozov et al., 2015), Sokoban puzzles (Kartal et al., 2016), and visual programming tasks (Ahmed et al., 2020; Ghosh et al., 2022). In particular, our work is related to (Ahmed et al., 2020; Ghosh et al., 2022) that proposed symbolic methods guided by hand-crafted constraints and Monte Carlo Tree Search to generate high-quality visual programming tasks. However, their symbolic methods still suffer from intractably large spaces of feasible tasks and codes for a given specification, and could take several minutes to generate an output task for an input specification; see Section 4 for comparison to our technique. In general, a major shortcoming of using purely symbolic generative methods in the above-mentioned works is that the generative process is typically time-inefficient and not suitable for applications that require real-time synthesis.

Against that backdrop, the main research question is: *Can we develop neuro-symbolic techniques that can synthesize high-quality visual programming tasks while being robust and efficient?* To this end, we develop NEURTASKSYN, a novel neuro-symbolic technique that can synthesize programming tasks for input specifications in the form of desired programming concepts exercised by its solution code and constraints on the visual task. Given a task synthesis specification as input (Figure 1c), NEURTASKSYN uses two components trained via reinforcement learning procedure: the first component generates possible solution codes, and the second component involves guiding an underlying symbolic execution engine that generates visual tasks for these codes. Our main results and contributions are summarized below: I. We formulate visual programming puzzle synthesis as a sequential decision-making process and propose NEURTASKSYN, a novel neuro-symbolic technique for synthesizing visual programming tasks (Section 3). II. We demonstrate the performance of NEURTASKSYN by comparing it to baselines from existing works (Section 4). III. We demonstrate the effectiveness of NEURTASKSYN through an extensive evaluation on task specifications from real-world programming platforms (Section 5). IV. We publicly release the implementation and datasets to facilitate future research.[1]

## 1.2 Related work

**Educational task generation.** Earlier works on task generation considered simpler domains such as algebra problems where solutions follow well-defined procedures. These works employed classical methods such as template-based (Polozov et al., 2015; Singh et al., 2012) or exhaustive enumeration-based task generation (Ahmed et al., 2020; Alvin et al., 2014). We aim to synthesize programming task given well-structured specifications, similar to previous works on visual task synthesis (Ahmed et al., 2020; Ghosh et al., 2022). While the existing works focused on offline generation, we seek to learn neural models that can generate tasks in real-time, as highlighted in Figure 1. Recent works have also explored the use of LLMs to generate new assignments for Python programming (Sarsa et al., 2022; Phung et al., 2023b) with mixed results.

**Spatial and logical reasoning for LLMs.** Recent studies have focused on assessing the various capabilities of LLMs (Bubeck et al., 2023; Arora et al., 2023; Bang et al., 2023). These works show that such state-of-the-art models have already achieved impressive generative capabilities for several programming education scenarios, including program repair and hint generation (Sarsa et al., 2022; Leinonen et al., 2023). However, these models still lack crucial capabilities like program execution, symbolic reasoning, and planning that are needed for structured task generation in visual programming domains (Valmeekam et al., 2022; Huang & Chang, 2022; Kaddour et al., 2023; Phung et al., 2023b). Based on (Huang et al., 2023), the LLMs' inherent capabilities alone are still insufficient for many general reasoning tasks and require the guidance of an external tool or a human.

---

[1]GitHub repository: `https://github.com/machine-teaching-group/tmlr2024_neurtasksyn`.

## 2   Problem Setup

**Visual programming tasks.** We define a task as a tuple $\mathtt{T} := (\mathtt{T}_{\mathrm{IO}}, \mathtt{T}_{\mathrm{code}})$, where $\mathtt{T}_{\mathrm{IO}}$ denotes the visual puzzle and $\mathtt{T}_{\mathrm{code}}$ denotes additional constraints on a solution code of this puzzle. This task space is general enough to encompass popular visual programming domains, including block-based programming domain of *Hour of Code:Maze Challenge* by Code.org (HoCMaze) (Code.org, 2013b;a) and text-based programming domain of Karel (Pattis et al., 1995). For instance, the task in Figure 1b is based on HoCMaze:MAZE20 task Code.org (2013a) – it is comprised of a visual puzzle where a solution code, when executed, will take the avatar to the goal without crashing into walls. Additional constraints on a solution code specify that it should use only blocks from the set {move, turnLeft, turnRight, REPEATUNTIL, IFELSE} and have a size $\leq 7$. We give a similar illustrative example for Karel in Section 5.

**Solution codes of a task.** We define the space of all possible codes $\mathbb{C}$ in a domain via a domain-specific language (DSL) (Gulwani et al., 2017). For instance, in our evaluation with HoCMaze and Karel programming domains, we will use their corresponding DSLs as introduced in (Bunel et al., 2018; Ahmed et al., 2020). For a given task $\mathtt{T}$, a code $\mathtt{C} \in \mathbb{C}$ is a solution code if the following holds: $\mathtt{C}$ successfully solves $\mathtt{T}_{\mathrm{IO}}$ while respecting $\mathtt{T}_{\mathrm{code}}$. For example, the codes in Figures 1b and 1d use only blocks from the set {move, turnLeft, turnRight, REPEATUNTIL, IFELSE}, have sizes equal to 7, and when executed, will take the avatar to the goal. Note that IFELSE is considered a single block.

**Task synthesis specification.** We now introduce a notation to specify desired tasks for synthesis that exercise certain programming concepts in their solution codes and respect certain constraints on the visual puzzle. We define a task synthesis specification as a tuple $\psi := (\psi_{\mathrm{IO}}, \psi_{\mathrm{sketch}}, \psi_{\Delta})$, where $\psi_{\mathrm{IO}}$ is a partially initialized visual puzzle, $\psi_{\mathrm{sketch}}$ is a code sketch (i.e., a partial code) capturing the structure that should be followed by the synthesized task's solution codes, and $\psi_{\Delta}$ specifies additional constraints on solution codes. For instance, the input task synthesis specification $\psi^{\mathrm{in}}$ in Figure 1c is extracted from $\mathtt{T}^{\mathrm{ref}}$ and $\mathtt{C}^{\mathrm{ref}}$ in Figure 1b – here, $\psi_{\mathrm{IO}}^{\mathrm{in}}$ is a 12x12 maze with certain cells initialized to free or wall cells; $\psi_{\mathrm{sketch}}^{\mathrm{in}}$ specifies the structure and some already initialized conditionals, loops and actions that should be used in solution codes; $\psi_{\Delta}^{\mathrm{in}}$ specifies constraints related to what kind of blocks can be used and the size of a solution code.

**Synthesis objective.** Given a task synthesis specification $\psi^{\mathrm{in}} := (\psi_{\mathrm{IO}}^{\mathrm{in}}, \psi_{\mathrm{sketch}}^{\mathrm{in}}, \psi_{\Delta}^{\mathrm{in}})$ as input, we seek to generate a task $\mathtt{T}^{\mathrm{out}} := (\mathtt{T}_{\mathrm{IO}}^{\mathrm{out}}, \mathtt{T}_{\mathrm{code}}^{\mathrm{out}})$ as output. Inspired by human-centered task quality criteria for the visual programming domains (Ahmed et al., 2020; Ghosh et al., 2022), we design objectives that capture the quality of desirable tasks. To formally set our synthesis objective and evaluation metrics, below we introduce different binary criteria (taking values 1 or 0) that we want $\mathtt{T}^{\mathrm{out}}$ to satisfy w.r.t. $\psi^{\mathrm{in}}$:

- **O1:Validity.** A task $\mathtt{T}^{\mathrm{out}}$ is valid (i.e., value 1) iff: (a) $\mathtt{T}_{\mathrm{IO}}^{\mathrm{out}}$ respects $\psi_{\mathrm{IO}}^{\mathrm{in}}$, (b) $\mathtt{T}_{\mathrm{code}}^{\mathrm{out}}$ respects $(\psi_{\mathrm{sketch}}^{\mathrm{in}}, \psi_{\Delta}^{\mathrm{in}})$, and (c) there exists at least one solution code $\mathtt{C} \in \mathbb{C}$ for $\mathtt{T}^{\mathrm{out}}$.

- **O2:Concepts.** A task $\mathtt{T}^{\mathrm{out}}$ conceptually captures specification $\psi^{\mathrm{in}}$ (i.e., value 1) iff: (a) there exists at least one solution code $\mathtt{C} \in \mathbb{C}$ that respects $\psi_{\mathrm{sketch}}^{\mathrm{in}}$ and (b) all solution codes $\mathtt{C} \in \mathbb{C}$ are at least of the complexity required by $\psi_{\mathrm{sketch}}^{\mathrm{in}}$. We define complexity w.r.t. the total number of programming constructs used (e.g., REPEATUNTIL, WHILE, IF, IFELSE) and the level of code nesting. We refer to the level of nesting as code depth in the remaining content of the paper.

- **O3:Trace.** A task $\mathtt{T}^{\mathrm{out}}$ captures the following synthesis property: for any solution $\mathtt{C}$ for $\mathtt{T}^{\mathrm{out}}$ that respects $(\psi_{\mathrm{sketch}}^{\mathrm{in}}, \psi_{\Delta}^{\mathrm{in}})$, the execution trace of $\mathtt{C}$ on $\mathtt{T}^{\mathrm{out}}$ executes each loop or conditional at least $n$ times. This property is inspired by real-world tasks that are easy to comprehend, as indicated by task quality criteria used in (Ahmed et al., 2020; Ghosh et al., 2022); we will use $n = 2$ in Section 5 evaluation.

- **O4:Overall.** This objective is 1 only if all the above objectives (O1, O2, O3) are satisfied for a task $\mathtt{T}^{\mathrm{out}}$.

- **O5:Human.** This objective captures the quality of a task $\mathtt{T}^{\mathrm{out}}$ from an expert's point of view. The assessed criteria are the conceptual correctness w.r.t. the input specification and the visual quality of the task. Based on these criteria, the expert reports an overall binary score.

# 3 Our Synthesis Technique NeurTaskSyn

In this section, we present NEURTASKSYN, our neuro-symbolic technique to synthesize visual programming tasks ($\mathtt{T}^{\mathrm{out}}$) for an input specification ($\psi^{\mathrm{in}}$).

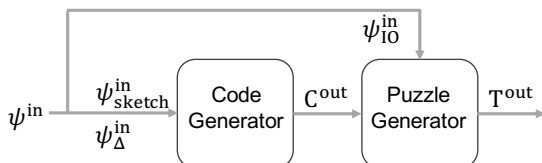

Figure 2: Components of task synthesis.

## 3.1 Overview

As noted in Section 1, a key challenge in synthesizing tasks is that the mapping from the space of visual tasks to their solution codes is highly discontinuous – a small modification in the output task could make it invalid or semantically incorrect w.r.t. the input specification (Ahmed et al., 2020). One way to tackle this challenge is to first reason about a possible solution code and then generate visual puzzles based on execution traces of this code (Gulwani, 2014; Kartal et al., 2016; Ahmed et al., 2020; Tercan et al., 2023). This motivates two components in our synthesis process shown in Figure 2: the first component generates possible solution codes $\mathtt{C}^{\mathrm{out}}$ (akin to that of program synthesis (Gulwani et al., 2017)); the second component generates the visual puzzle for these codes via symbolic execution (akin to the idea of test-case generation (King, 1976)). After synthesizing both, we can get all the elements of $\mathtt{T}^{\mathrm{out}}$. $\mathtt{T}^{\mathrm{out}}_{\mathrm{IO}}$ is based on the generated puzzle. $\mathtt{T}^{\mathrm{out}}_{\mathrm{code}}$ is extracted from the size and types of blocks of $\mathtt{C}^{\mathrm{out}}$.[2] Next, we discuss the components of NEURTASKSYN.

## 3.2 Generating the Solution Code $\mathtt{C}^{\mathbf{out}}$

The code generator component takes the elements of the specification, $(\psi^{\mathrm{in}}_{\mathrm{sketch}}, \psi^{\mathrm{in}}_{\Delta})$, that enforce constraints on solution codes of the desired task and accordingly generates a possible solution code $\mathtt{C}^{\mathrm{out}}$. Below, we briefly describe a base symbolic engine to generate syntactically valid codes from specifications via random search and then a neural model to guide this base engine; full details of this component are deferred to Appendix D.1.

**Base symbolic engine.** The base engine operates on Abstract Syntax Tree representation of code sketches (i.e., partial codes). The engine generates codes by sampling tokens (i.e., action blocks, conditions, and iterators) from underlying DSL while respecting specification. Even though this ensures that a generated code is syntactically correct and valid w.r.t. specification, it could have semantic irregularities. Such irregularities can lead to confusing tasks (e.g., see Figure 8c, where the IF and ELSE bodies are the same).

**Neural model.** The neural model is trained to guide the sampling process of the base symbolic engine. This neural model is akin to a program synthesizer, but it has a different objective. It aims to synthesize codes that are semantically correct and can lead to high-quality tasks. One could use a variety of architectures, for instance, transformer-based (Le et al., 2022; Fried et al., 2022; Li et al., 2022; Wang et al., 2021) or custom-made encoder-decoder approaches (Balog et al., 2017; Bunel et al., 2018; Yin & Neubig, 2017). In our work, we use an LSTM-based decoder (Hochreiter & Schmidhuber, 1997) because of their extensive use in the existing literature on generating program solutions for an input visual programming task (Devlin et al., 2017; Bunel et al., 2018; Shin et al., 2019; Gupta et al., 2020). In our setting, the input corresponds to the code specification. Similar to (Bunel et al., 2018), we use an imitation (supervised) learning approach to train the LSTM-based neural model, learning to predict in a token-by-token manner.

## 3.3 Generating the Visual Puzzle $\mathtt{T}^{\mathbf{out}}_{\mathbf{IO}}$

The puzzle generator component takes the element of the specification, $\psi^{\mathrm{in}}_{\mathrm{IO}}$, that enforces constraints on the visual puzzle along with generated code $\mathtt{C}^{\mathrm{out}}$ and accordingly generates a visual puzzle $\mathtt{T}^{\mathrm{out}}_{\mathrm{IO}}$. We first describe a base symbolic engine that performs symbolic execution of $\mathtt{C}^{\mathrm{out}}$ to generate a semantically valid puzzle via random search and then describe a neural model to guide this base engine. We then offer details regarding how we formulate the neurally guided symbolic engine as a reinforcement learning (RL) agent.

**Base symbolic engine.** The base engine performs symbolic execution of $\mathtt{C}^{\mathrm{out}}$ on $\psi^{\mathrm{in}}_{\mathrm{IO}}$ which has uninitialized elements/unknowns. This symbolic execution emulates an execution trace of $\mathtt{C}^{\mathrm{out}}$ and makes decisions about

---

[2]It is possible that code $\mathtt{C}^{\mathrm{out}}$ generated during intermediate step is not a solution for $\mathtt{T}^{\mathrm{out}}$, e.g., when $\mathtt{C}^{\mathrm{out}}$ is semantically incorrect and cannot generate a corresponding puzzle. In this case, we set the size constraints in $\mathtt{T}^{\mathrm{out}}_{\mathrm{code}}$ as specified by $\psi^{\mathrm{in}}_{\Delta}$.

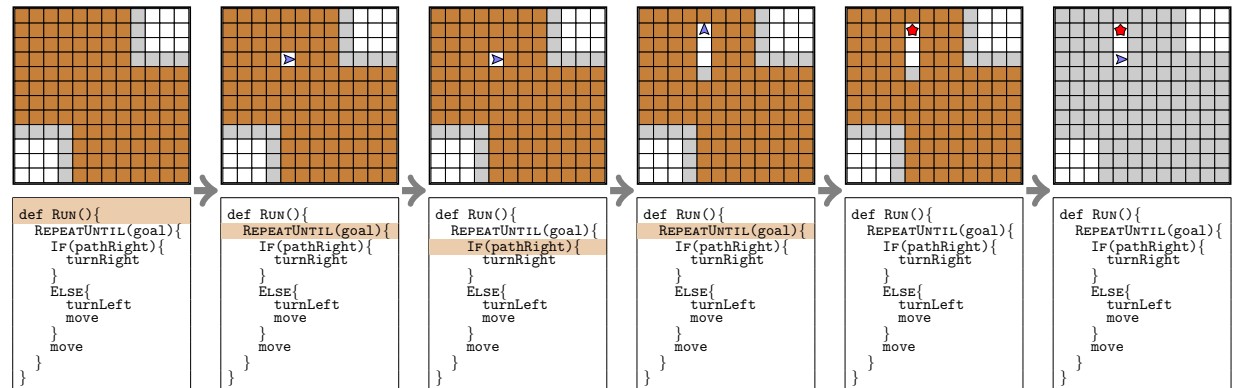

Figure 3: Illustration of symbolic execution. In the first step, the execution of $\texttt{C}^{\text{out}}$ starts on the partially initialized $\psi_{\text{IO}}^{\text{in}}$. In the second step, the agent's location and orientation are set and execution continues to REPEATUNTIL(goal), which is decided to false in this particular example. This decision does not immediately affect the visual grid. In the third step, the execution continues to IF(pathRight), which is decided to false and the avatar moves accordingly. The effect of these actions can be seen in the next state of the visual grid. In the fourth step, the execution continues to REPEATUNTIL(goal), which is decided to true, giving the goal location at step five. This leads to the sixth step, which is the end of the execution. The remaining unknowns are replaced with walls.

unknowns resulting in a concrete instantiation of $\psi_{\text{IO}}^{\text{in}}$ to $\texttt{T}_{\text{IO}}^{\text{out}}$. The outcome of these decisions affects the quality of the generated $\texttt{T}_{\text{IO}}^{\text{out}}$. Figure 3 shows an example of symbolic execution done by the symbolic engine. During this execution process, an emulator will replace unknowns with free cells, walls, the avatar position, and the goal location as indicated by the symbolic engine. The highlighted blocks of code in Figure 3 show the steps where the symbolic execution engine takes decisions, leading to one possible path in the execution of $\texttt{C}^{\text{out}}$ and an instantiation of $\texttt{T}_{\text{IO}}^{\text{out}}$. A code could have a potentially unbounded number of possible execution traces, and randomly taking decisions would typically lead to a lower quality task (King, 1976; Ahmed et al., 2020). Deciding the path of the execution maps to a sequential decision-making process. We aim to guide this decision-making process to achieve execution traces that lead to higher-quality tasks.

**Neural model.** We train the neural model to guide the decision-making process of the base symbolic engine. Our neural architecture uses a CNN-based encoder for partial visual puzzles and combines it with code execution-related features (e.g., coverage, currently executed code block). This choice is based on previous works on program synthesis for visual programming tasks (Bunel et al., 2018; Gupta et al., 2020). Other works have investigated the use of Monte Carlo Tree Search (MCTS) (Kocsis & Szepesvári, 2006) strategy to guide the symbolic execution for generating better puzzles with fewer resources (Kartal et al., 2016; Ahmed et al., 2020). However, these works used MCTS at inference time without any learned policy and could take several minutes to generate an output task for an input specification. Instead, a neural model makes use of learned experience to lead to high-quality puzzles.

**Neurally guided symbolic engine as an RL agent.** To speed up the generation process at inference, we train an RL agent (Sutton & Barto, 2018) whose reward is defined via a scoring function $\mathcal{F}_{\text{score}}$ that captures the quality of the generated visual puzzle for an input specification; this scoring function is similar in spirit to that used for MCTS in (Kartal et al., 2016; Ahmed et al., 2020). The agent's goal is to make decisions about the unknowns encountered when symbolically executing a code with the objective of generating high-quality puzzles. More concretely, we consider an episodic Markov Decision Process (MDP) defined as a tuple $M = (S, A, P, R, S_0)$, where an episode corresponds to a full symbolic execution of a code and the agent interacts with the environment over discrete time steps $t$. The elements of the MDP are defined as follows:

- A state $s \in S$ is given by current status of visual puzzle and the block being executed (as in Figure 3);

- A set of state-depended actions $A_s$ can either be the set of initialization actions (i.e., $\{\texttt{pos}_1, \texttt{pos}_2,..., \texttt{pos}_n\}$), or a set of binary decisions (i.e, $\{\texttt{true}, \texttt{false}\}$);

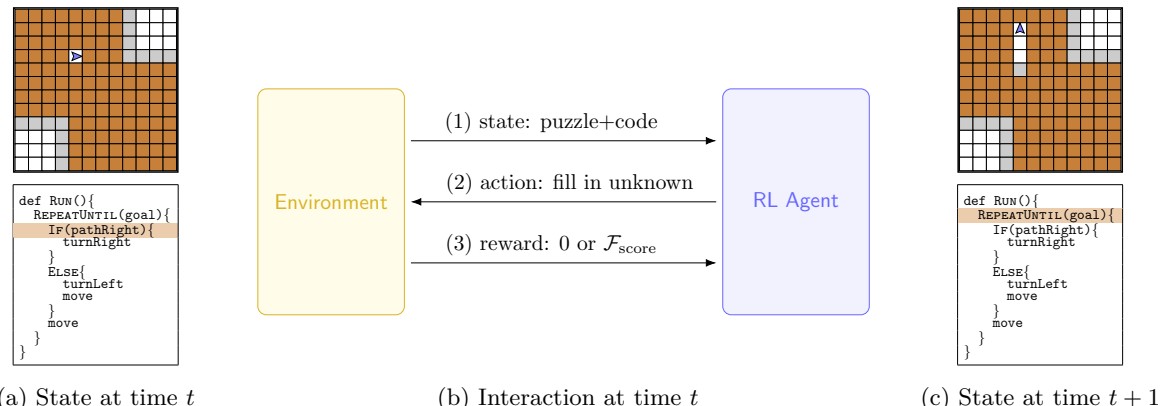

(a) State at time $t$         (b) Interaction at time $t$         (c) State at time $t+1$

Figure 4: Illustration showing the interaction between the RL agent and the environment. **(a)** shows state at time $t$, composed of code block being executed and status of the visual puzzle. **(b)** shows the interaction between the environment and the RL agent at time $t$. The environment comprises the code emulator and the scoring function. The RL agent is the neurally guided symbolic engine. **(c)** shows state at time $t+1$

- $P : S \times A \times S \to \mathbb{R}$ denotes deterministic transition dynamics, $P(s'|s,a) = 1$ for $s' = s \oplus a$, and 0 otherwise.

- $R : S \times A \to \mathbb{R}$ denotes a sparse reward. During the episode, the reward values are returned as 0; at the end of an episode, the reward values are computed using $\mathcal{F}_{\text{score}}(\texttt{T}^{\text{out}}, \texttt{C}^{\text{out}})$.

- $S_0 \subseteq S$ is the set of initial states. An initial state is composed of $\psi_{\text{IO}}^{\text{in}}$ and $\texttt{C}^{\text{out}}$.

In order to explain the interaction between the neurally guided symbolic engine and the emulator, we use a concrete example in Figure 4. Let us consider that the current state at time $t$ is composed of the block being executed (i.e., $\texttt{IF}(\texttt{pathRight})$ interrogation) and the status of the visual puzzle (i.e., the avatar surrounded by unknowns). This is depicted in Figure 4a. At this point, the neural model can guide the symbolic engine, as an interaction between the environment and the agent is expected (Figure 4b). The agent receives the current state of the emulator (i.e., as in Figure 4a) and outputs an action, i.e., the decision on how to fill in the unknown. In our case, the decision is that there is *no path to the right.* Then the environment computes a reward in accordance with the action taken by using $\mathcal{F}_{\text{score}}$. After the decision is taken, the emulator continues executing the upcoming blocks (i.e., $\texttt{turnLeft}$, $\texttt{move}$, $\texttt{move}$) until reaching the next step where a decision is needed (i.e., $\texttt{REPEATUNTIL}(\texttt{goal})$). The state at time $t+1$ is represented by the block being executed (i.e., $\texttt{REPEATUNTIL}(\texttt{goal})$) and the new status of the visual puzzle, as depicted in Figure 4c. The implementation details of the RL agent are provided in Appendix D.2.

## 4 Training and Validation on Synthetic Specifications

In this section, we train and validate NEURTASKSYN on synthetic datasets of task specifications. We employ different variants of NEURTASKSYN to quantify the utility of individual components and evaluate w.r.t. scoring and runtime metrics. Importantly, the models obtained here through training on synthetic dataset will be used for evaluation on real-world specifications in Section 5.

### 4.1 Evaluation Setup

**Visual programming domains.** We consider two popular visual programming domains: *Hour of Code:Maze Challenge* by Code.org (HoCMaze) (Code.org, 2013b;a) and Karel (Pattis et al., 1995), as introduced in Sections 1 and 2. Both these programming domains have been studied extensively in the literature on program/task synthesis (Bunel et al., 2018; Shin et al., 2019; Ahmed et al., 2020; Gupta et al., 2020) and computing education (Piech et al., 2015; Efremov et al., 2020; Ghosh et al., 2022). We define a few domain-specific elements for these visual programming domains. First, as introduced in Section 2, we use two domain-specific languages (DSLs) adapted from (Bunel et al., 2018; Ahmed et al., 2020). Second, as

mentioned in Section 3, we will use domain-specific scoring functions $\mathcal{F}_{\text{score}}^{\text{HoCMaze}}$ and $\mathcal{F}_{\text{score}}^{\text{Karel}}$ to capture quality of a visual programming task. In our work, we adapt scoring functions used in (Ahmed et al., 2020). Full details are provided in the supplementary material; in a nutshell, these scoring functions are designed to intuitively capture the synthesis objectives set in Section 2, including properties like code coverage and trace quality. These scoring functions will be used in different ways: (a) during training of NEURTASKSYN's puzzle generator as a reward for RL agent and during inference to select an output task from candidates; (b) when evaluating different techniques with a surrogate metric based on these scoring functions; (c) when computing the quality of expert-designed code-task pairs for the runtime experiments. For each domain, we will make use of an offline, time-intensive, method TASKORACLE(C): it does one million symbolic executions of a given code C and returns highest-scoring task w.r.t. scoring function $\mathcal{F}_{\text{score}}$ (cf. Footnote 3).

**Scoring-based metrics.** We introduce a binary success metric $\mathcal{M}(\psi^{\text{in}}, \text{T}^{\text{out}}, \text{C}^{\text{out}})$ that is used to compare the performance of different techniques and to pick hyperparameters. More concretely, $\mathcal{M}(\psi^{\text{in}}, \text{T}^{\text{out}}, \text{C}^{\text{out}})$ is 1 if the following hold: (i) the task $\text{T}^{\text{out}}$ is valid w.r.t. $\psi^{\text{in}}$; (ii) the generated code $\text{C}^{\text{out}}$ is semantically correct in a sense that it can lead to a valid task via TASKORACLE, i.e., $\mathcal{F}_{\text{score}}(\text{TASKORACLE}(\text{C}^{\text{out}}), \text{C}^{\text{out}}) > \lambda_1$; (iii) the generated task $\text{T}^{\text{out}}$ is good quality in comparison to the oracle-generated task, i.e., $\mathcal{F}_{\text{score}}(\text{T}^{\text{out}}, \text{C}^{\text{out}}) > \lambda_2 \cdot \mathcal{F}_{\text{score}}(\text{TASKORACLE}(\text{C}^{\text{out}}), \text{C}^{\text{out}})$. We use $\lambda_1 = 0$ and $\lambda_2 = 0.9$ in our experiments. For each technique, performance is computed as % success rate across the evaluation dataset w.r.t. $\mathcal{M}$; in total, we compute performance across three seeds and report averaged results as mean (stderr). Importantly, we note that this metric only serves as a surrogate metric for evaluation on synthetic dataset; the neural models trained here will be evaluated on real-world task specifications w.r.t. the synthesis objectives in the next section.

**Runtime metrics.** We aim to measure the resources needed for a technique to reach certain quality thresholds. We do this by first computing the score given by $\mathcal{F}_{\text{score}}$ to an expert-designed task-code pair. Then, we fix the code and measure the time and number of symbolic execution rollouts needed to reach a given % of the computed score. We evaluate each technique multiple times and report averaged results.

## 4.2 Evaluation w.r.t. Scoring-based Metrics

**Synthetic task specifications.** For training and evaluation of techniques, we create a dataset of synthetic task specifications per domain, referred to as $\mathbb{D} := \{\psi^{\text{in}}\}$. To create one specification $\psi^{\text{in}} := (\psi_{\text{sketch}}^{\text{in}}, \psi_{\text{IO}}^{\text{in}}, \psi_{\Delta}^{\text{in}})$, the most crucial part is getting a code sketch $\psi_{\text{sketch}}^{\text{in}}$ that respects the DSL and can lead to a valid code generation. We start by sampling a code $\text{C}^{\text{in}}$ from the DSL for a given structure, depth, and constructs – this sampling process is inspired by methods for synthetic dataset creation (Bunel et al., 2018; Shin et al., 2019; Ahmed et al., 2020). For each sampled code, we check its semantic validity, i.e., this code can lead to a high-quality task using TASKORACLE. Afterwards, for a sampled code $\text{C}^{\text{in}}$ we create its corresponding $\psi_{\text{sketch}}^{\text{in}}$ by keeping only the programming constructs (loops/conditionals) with a random subset of the booleans/iterators masked out. The rest of the $\psi^{\text{in}}$ elements are instantiated as follows: $\psi_{\text{IO}}^{\text{in}}$ is $16 \times 16$ size without any initialization, $\psi_{\Delta}^{\text{in}}$ enforces the underlying DSL and a randomly initialized size between the number of blocks in $\text{C}^{\text{in}}$ and 17. In our evaluation, we split $\mathbb{D}$ as follows: 80% for training the neural models ($\mathbb{D}^{\text{train}}$), 10% for calibration ($\mathbb{D}^{\text{cal}}$), and a fixed 10% for evaluation ($\mathbb{D}^{\text{test}}$).

**Techniques.** First we describe NEURTASKSYN, our main task synthesis technique from Section 3. For each domain (HoCMaze and Karel), we train a separate instance of NEURTASKSYN using the synthetic dataset introduced above. In Section 3, we described the generation process for a single "rollout", i.e., one $\text{C}^{\text{out}}$ and one puzzle $\text{T}_{\text{IO}}^{\text{out}}$ is generated. In practice, we use multiple rollouts to select a final output task $\text{T}^{\text{out}}$. More concretely, at inference time for a given $\psi^{\text{in}}$ as input, NEURTASKSYN generation process is captured by two parameters: number of code rollouts c by the code generator and number of puzzle rollouts p by the puzzle generator for each generated code. We denote these hyperparameters in subscript, e.g., NEURTASKSYN$_{\text{c:5,p:10}}$ for $5 \times 10$ rollouts. Out of these $c \times p$ candidates, the technique outputs one task $\text{T}^{\text{out}}$ along with solution code $\text{C}^{\text{out}}$ using its scoring function. Next, we describe different variants of NEURTASKSYN$_{\text{c,p}}$ and baselines:

- NEURPUZZLEGEN$_{\text{c:FIX,p}}$: This technique is a variant of NEURTASKSYN$_{\text{c,p}}$ to evaluate its puzzle generation component, assuming the code generator has access to code $\text{C}^{\text{in}}$ associated with specification $\psi^{\text{in}}$ in the dataset. At inference, NEURPUZZLEGEN$_{\text{c:FIX,p}}$ generation process is captured by hyperparameter p, i.e., number of puzzle rollouts. Out of p candidates, technique outputs one task, analogous to NEURTASKSYN.

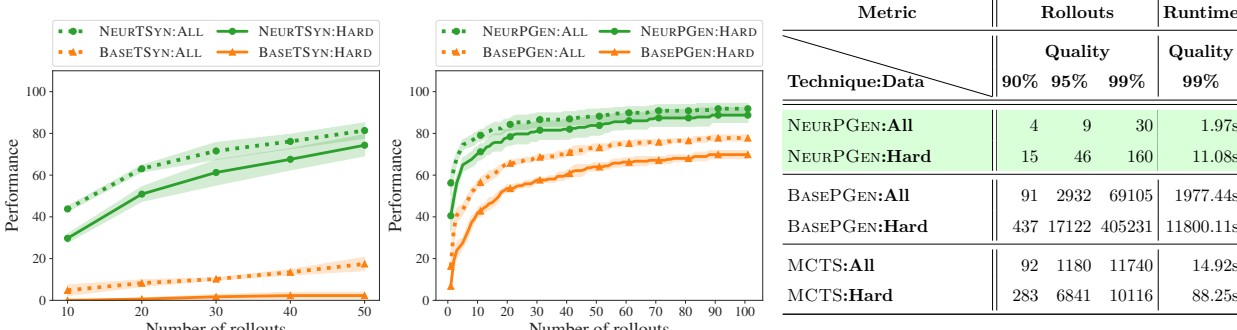

(a) BASETASKSYN, NEURTASKSYN  (b) BASEPUZZLEGEN, NEURPUZZLEGEN  (c) Results for runtime evaluation.

Figure 5: **(a)** Results for BASETASKSYN$_{c,p}$ and NEURTASKSYN$_{c,p}$ by increasing code rollouts c from 1 to 5 with fixed puzzle rollouts p = 10. **(b)** Results for BASEPUZZLEGEN$_{c:FIX,p}$ and NEURPUZZLEGEN$_{c:FIX,p}$ by increasing puzzle rollouts p from 1 to 100. **(c)** Table showing the runtime/rollouts comparison between different techniques, on tasks of different difficulty, when varying the required quality threshold. The quality threshold is expressed as % of the quality score of an expert-designed task. See further details in Section 4.

- BASETASKSYN$_{c,p}$ and BASEPUZZLEGEN$_{c:FIX,p}$: These techniques operate similar to NEURTASKSYN$_{c,p}$ and NEURPUZZLEGEN$_{c:FIX,p}$, but use only symbolic engine with random search.[3]

**Results.** Figures 5a and 5b report results as we vary the number of rollouts for different techniques, evaluated on the full dataset and on a "hard" segment of the dataset, where $\psi_{sketch}^{in}$ uses at least 2 constructs and has a depth of 3. In summary, these results demonstrate the utility of different components of NEURTASKSYN and how the synthesis quality improves as we increase the number of rollouts.

### 4.3 Evaluation w.r.t. Runtime Metrics

**Code samples for puzzle generation.** For evaluating the runtime of our technique, we pick a set of codes designed by experts as solutions for six representative tasks from both the HoCMaze and Karel domains. We select those tasks that are common among our evaluation in Section 5 (see Figure 6) and the work of (Ahmed et al., 2020). These tasks are derived from HoC:Maze8, HoC:Maze9, HoC:Maze13, HoC:Maze18 (Code.org, 2013b), and Karel:OurFirst, Karel:Diagonal (CodeHS, 2012b).

**Techniques.** We focus on evaluating the runtime performance of three techniques. NEURPUZZLEGEN$_{c:FIX,p}$ and BASEPUZZLEGEN$_{c:FIX,p}$ are equivalent to the techniques of Section 4.2. MCTS$_{c:FIX,p}$ is the puzzle generation method based on Monte Carlo tree search (MCTS) (Ahmed et al., 2020), introduced in Section 3.3. More specifically, we do not set a fixed number of puzzle rollouts p, but we measure the values of p needed to reach certain quality thresholds.

**Results.** We present our results in Figure 5c, averaged over all the selected tasks. Additionally, we define an expert code as "hard" when using at least 2 constructs and having a depth of 3. We separately report the results on HoC:Maze18, which is considered hard. It can be noticed that the unguided symbolic execution of BASEPUZZLEGEN$_{c:FIX}$ leads to large execution times when aiming to synthesize high-quality tasks. MCTS$_{c:FIX}$ has faster runtimes than BASEPUZZLEGEN$_{c:FIX}$, but it becomes less scalable when required to come up with high-quality tasks for the hard code. NEURPUZZLEGEN$_{c:FIX}$ maintains low execution times and number of rollouts for any of the required quality thresholds, even when faced with a hard code.

## 5  Experiments on Real-World Specifications

In this section, we evaluate our task synthesis technique NEURTASKSYN on real-world specifications.

---

[3]TASKORACLE(C) introduced above is BASEPUZZLEGEN with p = $10^6$ rollouts for a fixed code C.

| $\psi^{\text{in}}$ | $\psi_{\text{sketch}}^{\text{in}}$ structure | (depth, constructs) | $\psi_{\text{IO}}^{\text{in}}$ | $\psi_{\Delta}^{\text{in}}$ | Source |
|---|---|---|---|---|---|
| $\psi 0$ | {RUN {REPEAT}} | (2, 1) | 16x16 empty | HoCMaze, blocks $\leq 10$ | HoC:Maze9 Code.org (2013b) |
| $\psi 1$ | {RUN {REPEATUNTIL}} | (2, 1) | 16x16 empty | HoCMaze, blocks $\leq 10$ | HoC:Maze13 Code.org (2013b) |
| $\psi 2$ | {RUN {REPEAT; REPEAT}} | (2, 2) | 16x16 empty | HoCMaze, blocks $\leq 10$ | HoC:Maze8 Code.org (2013b) |
| $\psi 3$ | {RUN {REPEATUNTIL{IFELSE}}} | (3, 2) | 16x16 empty | HoCMaze, blocks $\leq 10$ | HoC:Maze18 Code.org (2013b) |
| $\psi 4$ | {RUN {REPEATUNTIL{IF; IF}}} | (3, 3) | 16x16 empty | HoCMaze, blocks $\leq 10$ | HoC:Maze20 Code.org (2013b) |
| $\psi 5$ | {RUN} | (1, 0) | 16x16 empty | Karel, blocks $\leq 10$ | Karel:OurFirst CodeHS (2012b) |
| $\psi 6$ | {RUN {WHILE}} | (2, 1) | 16x16 empty | Karel, blocks $\leq 10$ | Karel:Diagonal CodeHS (2012b) |
| $\psi 7$ | {RUN {WHILE; WHILE}} | (2, 2) | 16x16 empty | Karel, blocks $\leq 10$ | Karel:RowBack CodeHS (2012b) |
| $\psi 8$ | {RUN {WHILE{IF}}} | (3, 2) | 16x16 empty | Karel, blocks $\leq 10$ | Karel:Stairway CodeHS (2012b) |
| $\psi 9$ | {RUN {WHILE{REPEAT}}} | (3, 2) | 16x16 empty | Karel, blocks $\leq 10$ | Karel:CleanAll |

Figure 6: Real-world task specifications for HoCMaze and Karel; $\psi_{\text{sketch}}^{\text{in}}$ is shortened for brevity; $\psi_{\Delta}^{\text{in}}$ indicated as booleans and actions allowed by each domain, with the addition of maximum size.

**Real-world task specifications.** We use a set of 10 task specifications from HoCMaze and Karel domains, shown in Figure 6. These task specifications are inspired by their source tasks (see "Source" column) in the following sense: we create a specification $\psi^{\text{in}}$ for which the corresponding source task is a desired task as would be created by experts. Figure 8 shows illustration of task synthesis for a variant of $\psi 3$ (source as MAZE18 HoCMaze task) where we used 12x12 grid size with certain cells pre-initialized; analogously, Figure 9 shows illustration of task synthesis for a variant of $\psi 8$ (source as STAIRWAY Karel task) where we used 12x12 grid size.

**Synthesis quality metrics.** We evaluate techniques w.r.t. different metrics, each corresponding to a synthesis objective introduced in Section 2. Even though objectives O1–O4 are quantitative, it is challenging to fully automate their evaluation because it requires analyzing properties of different possible solution codes of a generated task. We manually did this evaluation when computing performance for each technique and metric. To capture objective O5, we assess the quality of synthesized tasks using human experts' annotations. Each expert independently evaluated all specifications and all techniques. Additionally, we report a binary metric of whether $\texttt{C}^{\text{out}}$ solves $\texttt{T}^{\text{out}}$ to provide insight into the robustness of the synthesis process. Results are reported as a mean over 10 specifications $\psi^{\text{in}}$ from Figure 6; we evaluate over three seeds and report averaged results as mean (stderr). In total, three experts (i.e., one per seed) with experience in visual programming provided annotations for O5.

**Techniques.** We evaluate NEURTASKSYN$_{\text{c:10,p:100}}$ with c = 10 and p = 100, i.e., total of c $\times$ p = 1000 rollouts (see Section 4). Next, we describe additional techniques evaluated:

- BASETASKSYN$_{\text{c:10,p:100}}$ operates similarly to NEURTASKSYN$_{\text{c:10,p:100}}$, but uses only base symbolic engine with random search without any neural guidance (see Section 4).

- GPT4TASKSYN-*converse* is based on OpenAI's GPT-4 (OpenAI, 2023b). It uses conversation-style prompts that involved human guidance to correct mistakes. More concretely, it is based on a two-stage process as shown in Figure 2 – we first ask GPT-4 to generate a code $\texttt{C}^{\text{out}}$ for $\psi^{\text{in}}$ and then ask it to generate a puzzle $\texttt{T}_{\text{IO}}^{\text{out}}$ that could be solved by $\texttt{C}^{\text{out}}$. The first stage comprised 5 separate queries to generate a $\texttt{C}^{\text{out}}$: each query started with an initial prompt and then follow-up prompts to fix any mistakes. The second stage comprised of another 5 separate queries to generate a puzzle $\texttt{T}_{\text{IO}}^{\text{out}}$: each query started with an initial prompt and then follow-up prompts to fix any mistakes. Once we get $\texttt{C}^{\text{out}}$ and $\texttt{T}_{\text{IO}}^{\text{out}}$, we set $\texttt{T}_{\text{code}}^{\text{out}}$ as for NEURTASKSYN. We defer the prompts to the supplementary material. This variant of GPT4TASKSYN was used to synthesize the tasks in Figures 8 and 9.

- GPT4TASKSYN-*fewshot* uses the same two-stage synthesis process, but employs few-shot examples without any follow-up conversations.

- EXPERT refers to expert-designed tasks. In our setup, EXPERT simply outputs a task $\texttt{T}^{\text{out}}$ based on the source task associated with input specification $\psi^{\text{in}}$; moreover it appropriately adjusts $\texttt{T}_{\text{IO}}^{\text{out}}$ to match $\psi_{\text{IO}}^{\text{in}}$ layout and sets $\texttt{T}_{\text{code}}^{\text{out}}$ as allowed by $(\psi_{\text{sketch}}^{\text{in}}, \psi_{\Delta}^{\text{in}})$ and the size of the minimal solution code.

| Technique | O1:Validity | O2:Concepts | O3:Trace | O4:Overall | O5:Human | C^out solves T^out |
|---|---|---|---|---|---|---|
| NEURTASKSYN$_{c:10,p:100}$ | 1.00 (0.00) | 0.83 (0.04) | 0.80 (0.00) | 0.80 (0.00) | 0.77 (0.11) | 1.00 (0.00) |
| BASETASKSYN$_{c:10,p:100}$ | 0.97 (0.04) | 0.37 (0.08) | 0.33 (0.04) | 0.33 (0.04) | 0.20 (0.07) | 0.50 (0.12) |
| GPT4TASKSYN-*converse* | 0.97 (0.04) | 0.57 (0.11) | 0.60 (0.07) | 0.43 (0.11) | 0.27 (0.08) | 0.33 (0.08) |
| GPT4TASKSYN-*fewshot* | 0.80 (0.07) | 0.37 (0.11) | 0.57 (0.11) | 0.33 (0.08) | 0.13 (0.11) | 0.27 (0.04) |
| EXPERT | 1.00 | 1.00 | 1.00 | 1.00 | 1.00 | 1.00 |

Figure 7: Results on real-world task specifications for HoCMaze and Karel in Figure 6; see Section 5. In this figure, EXPERT refers to expert-designed tasks.

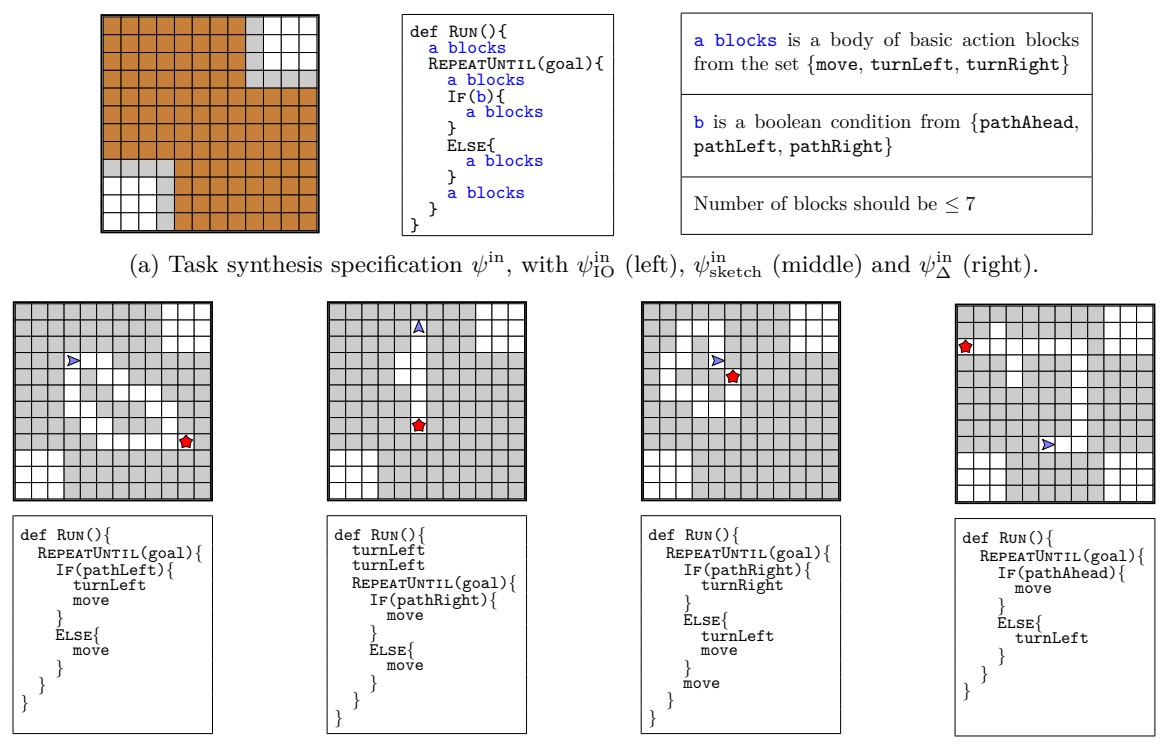

(a) Task synthesis specification $\psi^{\text{in}}$, with $\psi_{\text{IO}}^{\text{in}}$ (left), $\psi_{\text{sketch}}^{\text{in}}$ (middle) and $\psi_{\Delta}^{\text{in}}$ (right).

(b) $\text{T}_{\text{IO}}^{\text{out}}$, $\text{C}^{\text{out}}$ by GPT4TASKSYN  (c) $\text{T}_{\text{IO}}^{\text{out}}$, $\text{C}^{\text{out}}$ by BASETASKSYN  (d) $\text{T}_{\text{IO}}^{\text{out}}$, $\text{C}^{\text{out}}$ by NEURTASKSYN  (e) $\text{T}_{\text{IO}}^{\text{out}}$, $\text{C}^{\text{out}}$ by EXPERT

Figure 8: Illustrative example showcasing task synthesis inspired by the MAZE18 HoCMaze task Code.org (2013b;a). The given specification seeks to synthesize tasks where a solution code has the {REPEATUNTIL{IFELSE}} structure. **(a)** Task synthesis specification $\psi^{\text{in}} := (\psi_{\text{IO}}^{\text{in}}, \psi_{\text{sketch}}^{\text{in}}, \psi_{\Delta}^{\text{in}})$ is provided as input: $\psi_{\text{IO}}^{\text{in}}$ is a 12x12 maze with certain cells initialized to free (white) or wall (gray) cells, the rest being marked as unknowns (brown); $\psi_{\text{sketch}}^{\text{in}}$ along with $\psi_{\Delta}^{\text{in}}$ specify constraints on code solutions of a synthesized task. **(b–d)** show tasks $\text{T}^{\text{out}}$ and codes $\text{C}^{\text{out}}$ used as intermediate step to generate output tasks by three techniques. **(e)** shows task $\text{T}^{\text{out}}$ and code $\text{C}^{\text{out}}$ based on MAZE18. See Section 5.

**Results.** Figure 7 reports evaluation results for different techniques w.r.t. our task synthesis objectives. Next, we summarize some of our key findings. First, NEURTASKSYN has high performance of at least 0.77 across all metrics. The illustrative examples in Figures 8 and 9 showcase the high-quality of tasks synthesized by NEURTASKSYN, matching interesting characteristics of real-world tasks from EXPERT. Second, BASETASKSYN, GPT4TASKSYN-*converse*, and GPT4TASKSYN-*fewshot* struggle on objectives O2 and O3. Their low performance can be explained, in part, by failure to generate a valid task/code pair. The illustrative examples in Figures 8 and 9 further highlight the issues of tasks generated by these techniques. For instance, BASETASKSYN's $\text{T}^{\text{out}}$ in Figure 8c can be solved by a simpler code with lower depth than specified in the input specification; GPT4TASKSYN-*converse*'s $\text{T}^{\text{out}}$ in Figures 8b and 9b are not solvable by codes that would match the input specification. GPT4TASKSYN-*fewshot* shows that changing the prompting strategy does not help with increasing the performance. In summary, these results demonstrate the effectiveness of NEURTASKSYN in synthesizing high-quality visual programming tasks for real-world specifications and some

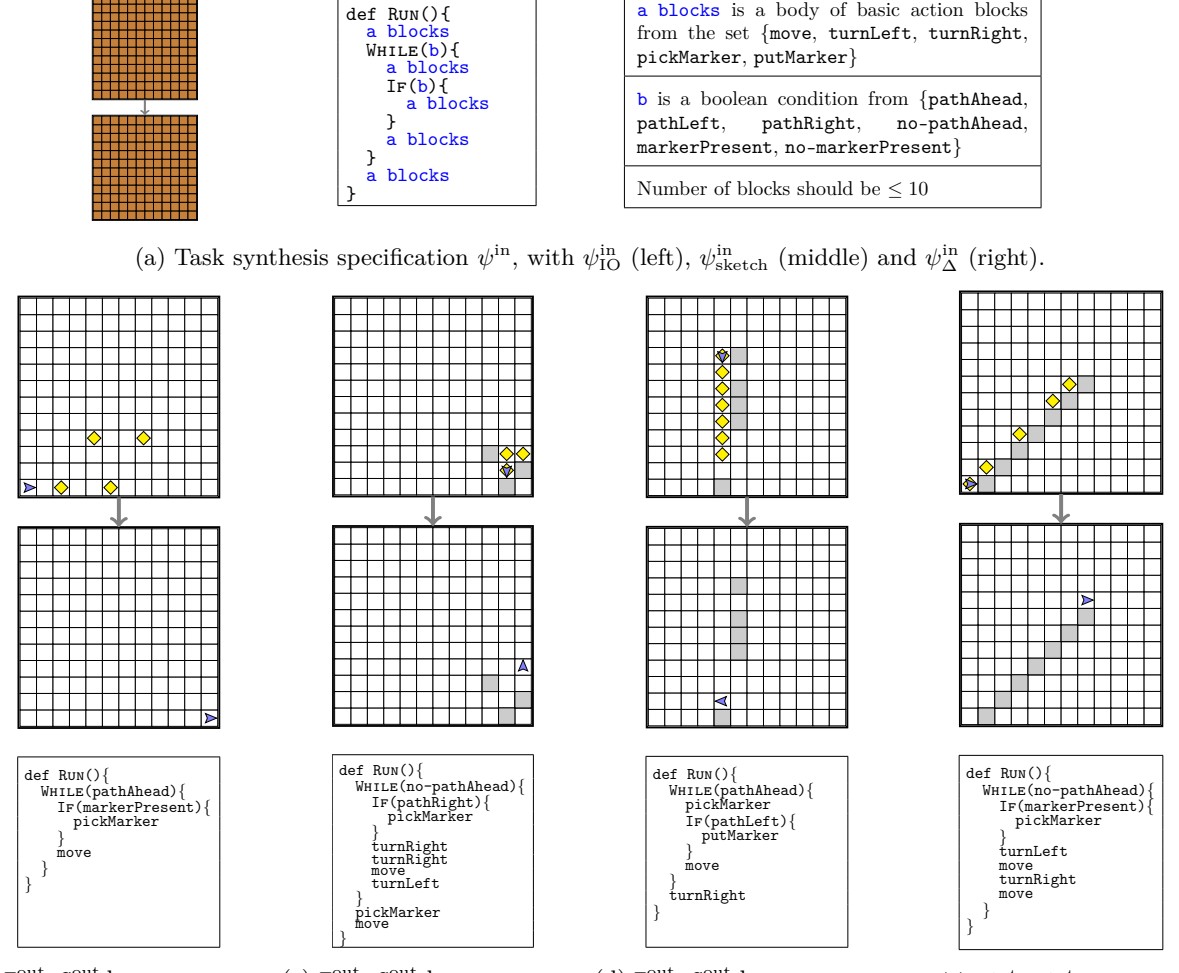

(a) Task synthesis specification $\psi^{\text{in}}$, with $\psi^{\text{in}}_{\text{IO}}$ (left), $\psi^{\text{in}}_{\text{sketch}}$ (middle) and $\psi^{\text{in}}_{\Delta}$ (right).

(b) $\text{T}^{\text{out}}_{\text{IO}}$, $\text{C}^{\text{out}}$ by GPT4TASKSYN  (c) $\text{T}^{\text{out}}_{\text{IO}}$, $\text{C}^{\text{out}}$ by BASETASKSYN  (d) $\text{T}^{\text{out}}_{\text{IO}}$, $\text{C}^{\text{out}}$ by NEURTASKSYN   (e) $\text{T}^{\text{out}}_{\text{IO}}$, $\text{C}^{\text{out}}$ by EXPERT

Figure 9: Illustrative example showcasing task synthesis inspired by the STAIRWAY Karel task Pattis et al. (1995); CodeHS (2012b;a). This specification seeks to synthesize tasks where a solution code has the $\{\text{WHILE}\{\text{IF}\}\}$ structure. **(a)** Task synthesis specification $\psi^{\text{in}} := (\psi^{\text{in}}_{\text{IO}}, \psi^{\text{in}}_{\text{sketch}}, \psi^{\text{in}}_{\Delta})$ is provided as input: $\psi^{\text{in}}_{\text{IO}}$ is a pregrid-postgrid pair with unitialized grid of size 12x12 ; $\psi^{\text{in}}_{\text{sketch}}$ along with $\psi^{\text{in}}_{\Delta}$ specify constraints on code solutions of a synthesized task. **(b–d)** show tasks $\text{T}^{\text{out}}$ and codes $\text{C}^{\text{out}}$ used as intermediate step to generate output tasks by three techniques. **(e)** shows task $\text{T}^{\text{out}}$ and code $\text{C}^{\text{out}}$ based on STAIRWAY. See Section 5.

of the challenges in synthesizing visual programming tasks by state-of-the-art neural generative models as the synthesis process requires logical, spatial, and programming skills.

## 6 Concluding Discussions

We developed a novel neuro-symbolic technique, NEURTASKSYN, that can synthesize visual programming tasks for a given specification. We demonstrated both the efficiency and effectiveness of NEURTASKSYN through an extensive evaluation on synthetically generated and reference tasks picked from popular visual programming environments. We believe our proposed technique has the potential to enhance introductory programming education by synthesizing personalized content. Moreover, we showcase the challenges that purely neural generative models and purely symbolic models face. BASETASKSYN, a purely symbolic model, can lead to semantic irregularities in codes and low-quality tasks. On the other hand, purely generative techniques based on GPT-4 struggle to generate high-quality puzzles. This result aligns with findings in contemporary studies that GPT-4 may face challenges in code execution, symbolic operations, test-case generation, and task synthesis (Bubeck et al., 2023; Phung et al., 2023b).

There are many interesting directions for future work. First, we have built our LSTM/CNN-based architecture specialized for task synthesis objectives; it would be interesting to fine-tune models like GPT-4 to improve its capabilities for synthesizing visual programming tasks. Moreover, we can analyze the logical and spatial abilities of these fine-tuned models by curating benchmarks in visual programming domains. Second, our methodology focused on visual programming; it would be interesting to develop generative models for synthesizing tasks in other programming domains, such as synthesizing Python problems that match a certain input-output configuration and other specifications regarding the possible solution codes. Third, our evaluation study considered various objectives capturing human-centered aspects along with expert annotations; in the future, it would also be useful to conduct studies with human learners to evaluate the quality in terms of perceived difficulty or interpretability of synthesized tasks.

### Acknowledgements

Funded/Co-funded by the European Union (ERC, TOPS, 101039090). Views and opinions expressed are however those of the author(s) only and do not necessarily reflect those of the European Union or the European Research Council. Neither the European Union nor the granting authority can be held responsible for them.

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

## A  Table of Contents

In this section, we provide a brief description of the content provided in the appendices of the paper.

- Appendix B provides a discussion of the broader impact of our work and compute resources used.

- Appendix C presents the details about the generation of the illustrative examples from Figures 8 and 9, and shows their relationship with metrics O1-O4 described in Section 2.

- Appendix D gives more insights into the architecture described in Section 3.

- Appendix E provides additional details about results, the scoring functions, the synthetic dataset creation process, and the training process in Section 4.

- Appendix F provides the source task/code pairs used for creating the real-world task specifications in Section 5. It also provides more insights into the interaction with GPT-4.

## B  Discussion

**Broader impact.** This paper develops new techniques which have the potential of being used for improving pedagogy in visual programming environments. On the existing platforms, content is hand-curated by tutors, offering limited resources for students to practice on. We aim to tackle this challenge by synthesizing novel practice tasks that match a desired level of difficulty with regard to exercised content for a student. We believe our proposed technique has the potential to drastically enhance introductory programming education by synthesizing personalized content for students.

**Compute resources.** All the experiments were conducted on a cluster of machines equipped with Intel Xeon Gold 6142 CPUs running at a frequency of 2.60GHz.

# C Illustrative Examples: Details

In this section, we discuss the details regarding the generation and scoring for each of the techniques' output in Figure 8 for HoCMaze and Figure 9 for Karel.

## C.1 Example for HoCMaze in Figure 8

We present $\mathtt{T}^{\mathrm{out}}$, along with $\mathtt{C}^{\mathrm{out}}$ for each of the techniques with $\psi^{\mathrm{in}}$ as input in Figure 11; this figure expands on Figure 8 with additional details. We give additional explanations regarding how each of the techniques' output respects or not metrics O1-O3 (see Sections 2 and 5) in Figure 10.

**Generation/adjustment for GPT4TaskSyn in Figure 8.** When querying GPT-4 for this example, we set $\psi^{\mathrm{in}}_{\mathrm{IO}}$ as an empty 8x8 grid. We then expand the generated grid to a 12x12 grid and manually integrate the pattern seen in Figure 11a to match the specification. We discuss the results obtained by GPT4TaskSyn-*converse*.

**Generation/adjustment for BaseTaskSyn and NeurTaskSyn in Figure 8.** The neural model for puzzle generation is trained on 16x16 grids, yet the symbolic engine can support the existence of pre-initialized grids. Thus, we mask the upper-left part of the grid (4 rows and 4 columns), obtaining the 12x12 workspace for the technique. On top of that, on the remaining 12x12 grid, we pre-initialize the pattern seen in Figure 11a (i.e., lower-left and upper-right bounded squares).

**Generation/adjustment for Expert in Figure 8.** The output of EXPERT represents a manual adaptation of the HoC:Maze18 task to expand it to a 12x12 grid and to integrate the pattern seen in in Figure 8a.

| Technique | O1:Validity | O2:Concepts | O3:Trace | O4:Overall | $\mathtt{C}^{\mathrm{out}}$ solves $\mathtt{T}^{\mathrm{out}}$ |
|---|---|---|---|---|---|
| GPT4TaskSyn | 1 | 0 | 0 | 0 | 0 |
| BaseTaskSyn | 1 | 0 | 0 | 0 | 1 |
| NeurTaskSyn | 1 | 1 | 1 | 1 | 1 |
| Expert | 1 | 1 | 1 | 1 | 1 |

Figure 10: Scores showing whether the output $\mathtt{T}^{\mathrm{out}}$ for $\psi^{\mathrm{in}}$ of each technique respects the four metrics O1-O4, with the additional $\mathtt{C}^{\mathrm{out}}$ solves $\mathtt{T}^{\mathrm{out}}$ metric, for this HoCMaze example.

We provide explanations for each 0 entry for the objectives O1-O3 in Figure 10:

- GPT4TaskSyn for Objective O2: The only possible solution code $\mathtt{C}$ has depth 4 and uses 3 constructs (nested IfElse is needed), i.e., $\psi^{\mathrm{in}}_{\mathrm{sketch}}$ is not respected, hence O2 is 0.

- GPT4TaskSyn for Objective O3: There is no solution $\mathtt{C}$ that respects $\psi^{\mathrm{in}}_{\mathrm{sketch}}$, hence O3 is 0 by definition.

- GPT4TaskSyn for $\mathtt{C}^{\mathrm{out}}$ solves $\mathtt{T}^{\mathrm{out}}$: $\mathtt{C}^{\mathrm{out}}$, when executed on $\mathtt{T}^{\mathrm{out}}_{\mathrm{IO}}$, makes the avatar crash into a wall, hence $\mathtt{C}^{\mathrm{out}}$ is not a solution for $\mathtt{T}^{\mathrm{out}}$

- BaseTaskSyn for Objective O2: The IfElse block employed by $\mathtt{C}^{\mathrm{out}}$ is not required. This implies that there is a solution code $\mathtt{C}$ which has a lower complexity, i.e., a smaller depth and less constructs than required by $\psi^{\mathrm{in}}_{\mathrm{sketch}}$, hence O2 is 0.

- BaseTaskSyn for Objective O3: As the employed IfElse block is not required, it is possible to design a solution code that uses IfElse with a different conditional (e.g., If(pathLeft)Else) for which the body would never be executed, hence O3 is 0.

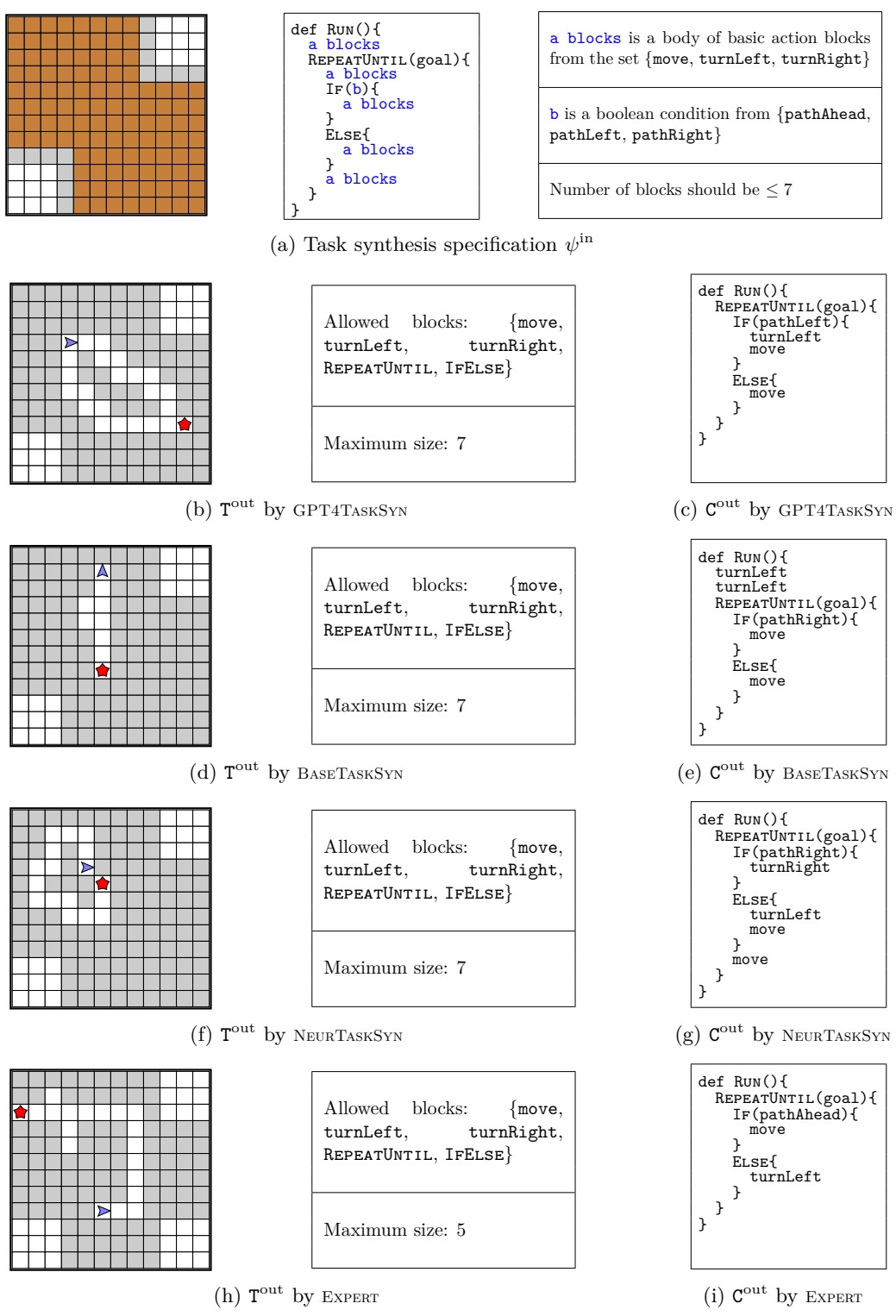

(a) Task synthesis specification $\psi^{\text{in}}$

(b) T$^{\text{out}}$ by GPT4TASKSYN

(c) C$^{\text{out}}$ by GPT4TASKSYN

(d) T$^{\text{out}}$ by BASETASKSYN

(e) C$^{\text{out}}$ by BASETASKSYN

(f) T$^{\text{out}}$ by NEURTASKSYN

(g) C$^{\text{out}}$ by NEURTASKSYN

(h) T$^{\text{out}}$ by EXPERT

(i) C$^{\text{out}}$ by EXPERT

Figure 11: Illustration containing the tuple T$^{\text{out}}$ for each technique, along with C$^{\text{out}}$, for this HoCMaze example.

### C.2 Example for Karel in Figure 9

We present $\mathtt{T}^{\mathrm{out}}$, along with $\mathtt{C}^{\mathrm{out}}$ for each of the techniques with $\psi^{\mathrm{in}}$ as input in Figure 13; this figure expands on Figure 9 with additional details. We give additional explanations regarding how each of the techniques' output respects or not metrics O1-O3 (see Sections 2 and 5) in Figure 12.

**Generation/adjustment for GPT4TaskSyn in Figure 9.** For this example, we set $\psi^{\mathrm{in}}_{\mathrm{IO}}$ as an empty 12x12 grid and query GPT-4. We discuss the results obtained by GPT4TASKSYN-*converse*.

**Generation/adjustment for BaseTaskSyn and NeurTaskSyn in Figure 9.** The neural model for puzzle generation is trained on 16x16 grids, yet the symbolic engine can support the existence of pre-initialized grids. Thus, we mask the upper-left part of the grid (4 rows and 4 columns), obtaining the 12x12 workspace for the technique.

**Generation/adjustment for Expert in Figure 9.** The output of EXPERT for this example is based on the Karel:Stairway task.

| Technique | O1:Validity | O2:Concepts | O3:Trace | O4:Overall | $\mathtt{C}^{\mathrm{out}}$ solves $\mathtt{T}^{\mathrm{out}}$ |
|---|---|---|---|---|---|
| GPT4TASKSYN | 0 | 0 | 0 | 0 | 0 |
| BASETASKSYN | 1 | 0 | 1 | 0 | 1 |
| NEURTASKSYN | 1 | 1 | 1 | 1 | 1 |
| EXPERT | 1 | 1 | 1 | 1 | 1 |

Figure 12: Scores showing whether the output $\mathtt{T}^{\mathrm{out}}$ for $\psi^{\mathrm{in}}$ of each technique respects the six metrics O1-O4, with the additional $\mathtt{C}^{\mathrm{out}}$ solves $\mathtt{T}^{\mathrm{out}}$ metric, for this Karel example.

We provide explanations for each 0 entry for the objectives O1-O3 in Figure 12:

- GPT4TASKSYN for Objective O1: $\mathtt{T}^{\mathrm{out}}_{\mathrm{IO}}$ cannot be solved with any code respecting $\mathtt{T}^{\mathrm{out}}_{\mathrm{code}}$, hence O1 is 0.

- GPT4TASKSYN for Objective O2: $\mathtt{T}^{\mathrm{out}}_{\mathrm{IO}}$ cannot be solved with any code respecting $\mathtt{T}^{\mathrm{out}}_{\mathrm{code}}$, hence O2 is 0.

- GPT4TASKSYN for Objective O3: $\mathtt{T}^{\mathrm{out}}_{\mathrm{IO}}$ cannot be solved with any code respecting $\mathtt{T}^{\mathrm{out}}_{\mathrm{code}}$, hence O3 is 0.

- GPT4TASKSYN for $\mathtt{C}^{\mathrm{out}}$ solves $\mathtt{T}^{\mathrm{out}}$: The generated code $\mathtt{C}^{\mathrm{out}}$ does not solve $\mathtt{T}^{\mathrm{out}}$, i.e., the pregrid is not transformed into the postgrid after code execution.

- BASETASKSYN for Objective O2: The employed $\mathtt{IF}$ block is not required. This implies that there is a solution code $\mathtt{C}$ which has a lower complexity, i.e., a smaller depth and less constructs than required by $\psi^{\mathrm{in}}_{\mathrm{sketch}}$, hence O2 is 0.

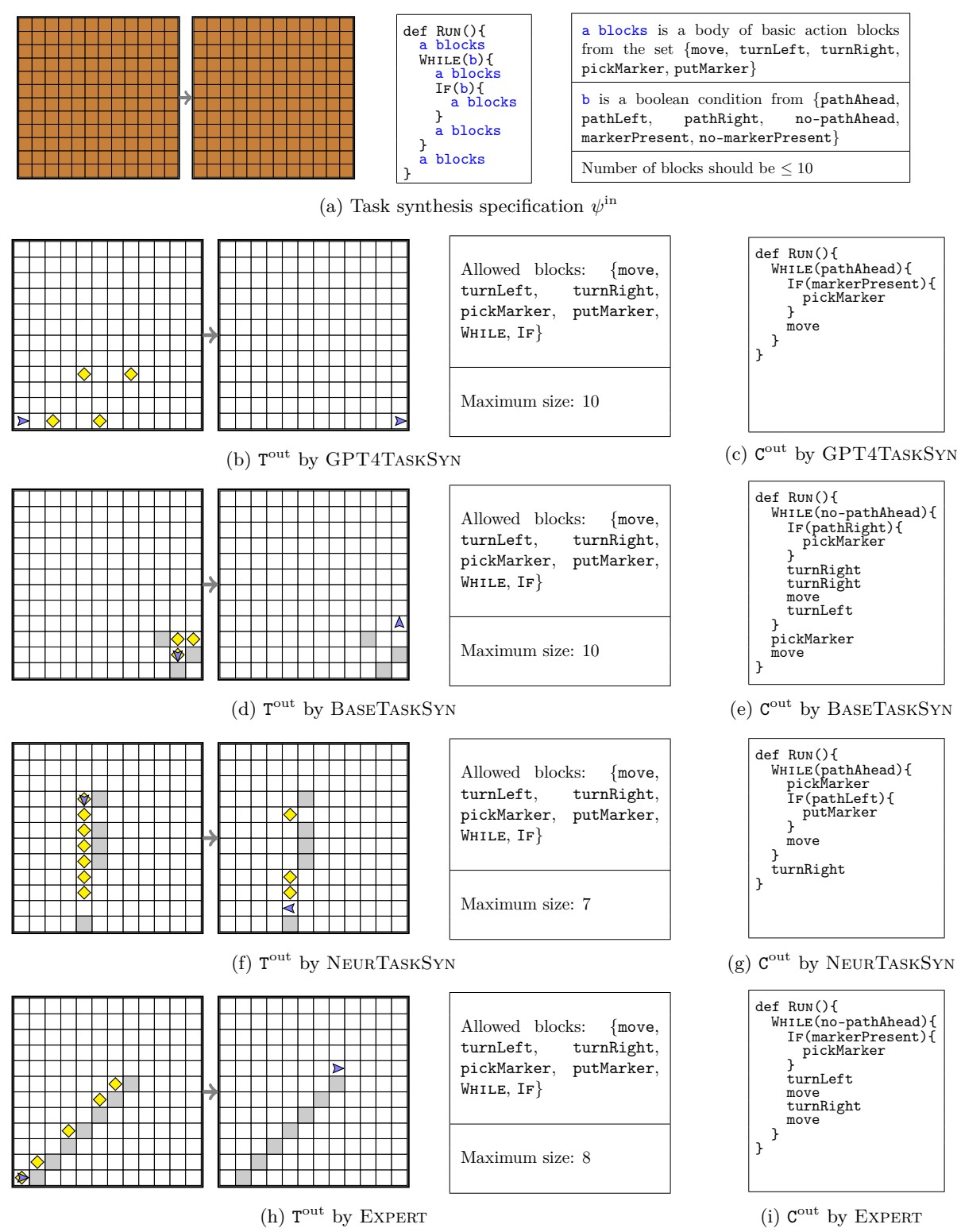

Figure 13: Illustration containing the tuple $\mathtt{T}^{\text{out}}$ for each technique, along with $\mathtt{C}^{\text{out}}$, for this Karel example.

# D   Our Synthesis Technique NeurTaskSyn: Details

Next, we give additional details regarding each module of our architecture. We present the interaction between the neural models and the underlying symbolic engines, the neural architecture, and the training procedures.

## D.1   Generating the Solution Code C^out

**Code generator visualization.** We describe the interaction between the neural model and the underlying symbolic engine for the code generator. For better understanding, we use one concrete example, illustrated in Figure 14. We consider the AST at time $t$ as presented in Figure 14a, where the previously taken decision was the addition of the `turnLeft` token. The symbolic engine continues its depth-first traversal of the AST and now needs to take the next decision for the subsequent '`a blocks`'. This is achieved by interrogating the neural component; interaction demonstrated in Figure 14b. We introduce the notion of a *budget*, which represents the number of available blocks that can be added to the AST so that the size specified by $\psi_\Delta^{\text{in}}$ is respected; in our example, the remaining budget is 2. It is passed as input for the neural model at time $t$. The neural model, based on the budget and its internal state, which keeps track of the previously taken decisions, outputs a logit for each decision, i.e., a set of logits $L_{\text{dict}}$. The symbolic engine accepts $L_{\text{dict}}$ and masks them according to the rules in the DSL, thus obtaining $L_{\text{dict}}^{\text{masked}}$. In our example, the only values in $L_{\text{dict}}^{\text{masked}}$ that are not masked are those of basic action blocks (i.e., `move`, `turnLeft`, `turnRight`) and the token that represents the end of the **Else** body. After mapping the logits to a probability distribution, the symbolic engine proceeds to sample a decision from it. In our example, `move` is sampled. The decision is passed to the neural model to update its internal state. The symbolic engine then updates the AST with the taken decision (i.e., `move`), thus obtaining the updated version of the AST for the next step at time $t + 1$, illustrated in Figure 14c. We generalize this process to every decision that needs to be taken while traversing the AST.

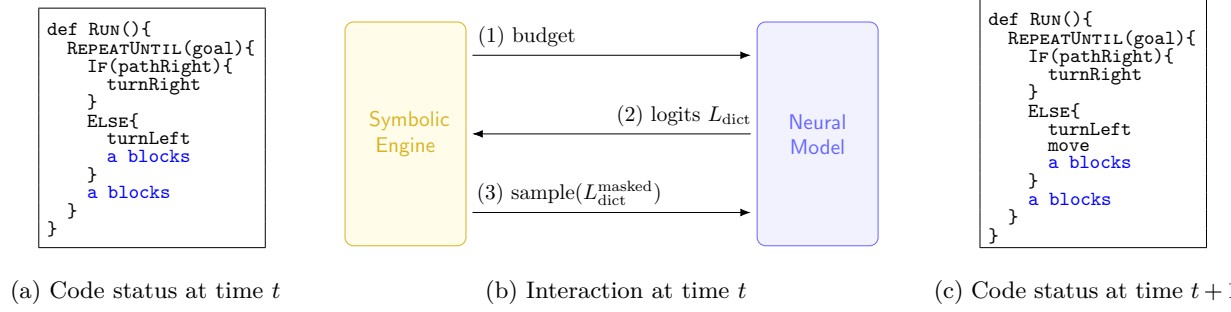

(a) Code status at time $t$        (b) Interaction at time $t$        (c) Code status at time $t+1$

Figure 14: Visualization of the interaction process between the neural model and the symbolic engine in the code generator component of NEURTASKSYN. (a) shows the AST at time $t$, where the first '`a blocks`' needs to be decided. (b) shows the interaction between the symbolic engine and the neural model at time $t$, where the symbolic engine first passes the *budget* (available blocks) to the neural model, the neural model computes the logits for all the tokens in the dictionary $L_{\text{dict}}$ and passes them back to the symbolic engine, which finally masks them obtaining $L_{\text{dict}}^{\text{masked}}$, applies softmax to obtain a probability and samples the next action, sending it to the neural model. (c) shows the AST at time $t + 1$, where the sampled `move` was integrated.

**Imitation learning procedure.** We will now give details about the learning procedure we used for the neural model. Given the fact that dataset $\mathbb{D} := \{\psi^{\text{in}}\}$ is accompanied by example codes, i.e., C^in for each $\psi^{\text{in}}$, we employ an imitation (supervised) learning approach, similar to (Devlin et al., 2017; Bunel et al., 2018). Thus, for each decision, we compute the cross-entropy with respect to the target decision. We force the agent to take the target decision afterward so the generated code does not digress from our example code.

**Neural architecture.** Here, we present in detail the architecture of the neural model we employ for code generation. Similar to (Bunel et al., 2018), we employ an LSTM-based (Hochreiter & Schmidhuber, 1997) recurrent neural network. We first convert code tokens to indexes based on a dictionary, then we pass them through an embedding layer. We do the same with the numeric representation of the *budget* (introduced previously). We concatenate both embeddings and pass them through a two-layer LSTM. Last, we convert the output of the LSTM to logits for each entry in the dictionary using a linear layer. The architecture can be observed in Figure 15.

| Input | Code token categorical (0-58) | Budget ordinal (0-16) |
|---|---|---|
| Embedding | Size = 256 | Size = 16 |
| LSTM 1 | Hidden dim = 256 | |
| LSTM 2 | Hidden dim = 256 | |
| Linear | Hidden dim $\times$ Dict size = 256 $\times$ 59 | |

Figure 15: Architecture of the neural model used by the code generator.

## D.2 Generating the Visual Puzzle $\mathsf{T}^{\mathbf{out}}_{\mathsf{IO}}$

**Reinforcement learning procedure.** We describe further details necessary for training our reinforcement learning (RL) agent. To learn a policy, we use policy gradient methods. These methods generally learn by using gradient ascent, thus updating the parameters $\theta$ of the parameterized policy $\pi_\theta(a|s)$ to increase the expected reward of the policy in the MDP. Naturally, a neural network can be used to learn the policy, where $\theta$ represents the network's weights. The network would take action $a$ and state $s$ as input, outputting a logit $H_\theta(a|s)$. Given the logits, we map them to a probabilistic distribution using softmax: $\pi_\theta(a|s) = \frac{\exp\left(H_\theta(a|s)\right)}{\sum_{a' \in A_s} \exp\left(H_\theta(a'|s)\right)}$. We use an actor-critic policy gradient method for agent training (Sutton & Barto, 2018). We denote with $\hat{v}(s,w)$ the value for state $s$ predicted by the critic with parameters $w$. As we operate on batches, the parameters of both the actor and the critic remain unchanged until a buffer is filled with a fixed number of episodes. Thus, for an initial state $s_0$ (i.e., an empty or pre-initialized task and a code, with the emulator reset), we execute the existing policy $\pi_\theta$ until the buffer is filled, generating several sequences of experience as tuples $(s_t, a_t, r_t)_{t=0..T}$, where $T$ represents a variable episode length. Thus, the losses for an episode are computed as a sum over the timesteps $t \in [0, T]$ as follows, for the actor (Equation 1) and for the critic (Equation 2, employing the smooth L1 loss, denoted as $\mathrm{L1}_{\mathrm{smooth}}$):

$$\mathrm{Loss}_\theta = \sum_{t=0}^{T} \left( \sum_{\tau=t}^{T} r_\tau - \hat{v}(s_t, w) \right) \cdot \nabla_\theta \log(\pi_\theta(a_t|s_t)) \tag{1}$$

$$\mathrm{Loss}_w = \sum_{t=0}^{T} \mathrm{L1}_{\mathrm{smooth}} \left( \sum_{\tau=t}^{T} r_\tau, \ \hat{v}(s_t, w) \right) \tag{2}$$

Finally, $\theta$ and $w$ are updated by using the computed losses over the entire batch, multiplied with a learning rate.

**Neural architecture.** We describe the architecture of the CNN-based neural model used by the puzzle generator. We employ a similar architecture for both the HoCMaze and Karel domains, as presented in Figure 16. Only the input size for the grid (i.e., $D \times 16 \times 16$, where $D = 12$ for HoCMaze and $D = 14$ for Karel) and code features (i.e., F, where $F = 9$ for HoCMaze and $F = 12$ for Karel) differ. We process the grid by 3 CNN blocks (i.e., one block composed of Conv2D, ReLU, and MaxPooling2D layers), after which we apply 5 fully connected (linear) layers, thus obtaining the grid embedding. To the grid embedding, we concatenate the code features, which are represented in a binary manner (e.g., increase in coverage, current decision type). We then pass the concatenated tensor through an additional fully-connected layer, and its output is then passed to both the action head and the value head (i.e., necessary for the Actor-Critic algorithm).

| Input | D × 16 × 16 Grid | Code features, Size = F |
|---|---|---|
| CNN Block 1 | Conv2D, kernel size = 3, padding 1, 64 × 64
ReLU
MaxPool2D, kernel size = 2, padding 0 | |
| CNN Block 2 | Conv2D, kernel size = 3, padding 1, 64 × 64
ReLU
MaxPool2D, kernel size = 2, padding 0 | |
| CNN Block 3 | Conv2D, kernel size = 3, padding 1, 64 × 64
ReLU
MaxPool2D, kernel size = 2, padding 0 | |
| Linear 1 | CNN output size (256) × 1024 | |
| Linear 2 | 1024 × 512 | |
| Linear 3 | 512 × 256 | |
| Linear 4 | 256 × 128 | |
| Linear 5 | 128 × 32 | |
| Linear 6 | Concatenated features size (32 + F) × 8 | |
| Linear 7 (Action and Value) | 8 × action space | 8 × 1 |

Figure 16: Architecture of the neural model used by the puzzle generator. D denotes the depth of the input grid and F denotes the size of the code features tensor, both different for each of the HoCMaze and Karel domains.

# E  Experimental Evaluation with Synthetic Task Specifications: Details

In this section, we detail the instantiations of the scoring functions, give more insight into the synthetic dataset creation process, and show the details of the training processes for both the code generator and the task generator.

## E.1  Scoring Function $\mathcal{F}_{\text{score}}$

Next, we describe the two instantiations for $\mathcal{F}_{\text{score}}$ as used in the two domains HoCMaze and Karel. We adopt a scoring function similar to that of (Ahmed et al., 2020), where $\mathcal{F}_{\text{score}}$ is used for guiding a Monte Carlo Tree Search, as an evaluation function that describes the desired properties of their system's output. We note that our method can work with any other instantiation of the scoring function $\mathcal{F}_{\text{score}}$. The instantiations we use for $\mathcal{F}_{\text{score}}$ for each of the HoCMaze and Karel domains are defined in Equations 3 and 4 and are comprised of different components: (i) $\mathcal{F}_{\text{cov}}(\texttt{T}_{\text{IO}}^{\text{out}}, \texttt{C}^{\text{out}}) \in [0,1]$ computes the coverage ratio, i.e., ratio of executed blocks to total number of blocks; (ii) $\mathcal{F}_{\text{sol}}(\texttt{T}_{\text{IO}}^{\text{out}}, \texttt{C}^{\text{out}}) \in \{0,1\}$ evaluates to 1 if $\texttt{C}^{\text{out}}$ correctly solves $\texttt{T}_{\text{IO}}^{\text{out}}$, i.e., no crashing, reaching the goal/converting the pre-grid to the post-grid; (iii) $\mathcal{F}_{\text{nocross}}(\texttt{T}_{\text{IO}}^{\text{out}}, \texttt{C}^{\text{out}}) \in [0,1]$ computes the ratio of cells visited exactly once with regard to the total number of visited cells; (iv) $\mathcal{F}_{\text{nocut}}(\texttt{T}_{\text{IO}}^{\text{out}}, \texttt{C}^{\text{out}}) \in \{0,1\}$ evaluates to 0 if there is a shortcut sequence comprised of basic actions; (v) $\mathcal{F}_{\text{notred}}(\texttt{T}_{\text{IO}}^{\text{out}}, \texttt{C}^{\text{out}}) \in \{0,1\}$ evaluates to 0 if there are redundant action sequences in $\texttt{C}^{\text{out}}$, e.g., sequences like $\texttt{turnLeft}$, $\texttt{turnRight}$, or if the codes obtained by eliminating one action, loop or conditional from $\texttt{C}^{\text{out}}$ solves $\texttt{T}_{\text{IO}}^{\text{out}}$; (vi) $\mathcal{F}_{\text{qual}}(\texttt{T}_{\text{IO}}^{\text{out}}, \texttt{C}^{\text{out}}) \in [0,1]$ evaluates the visual quality of $\texttt{T}_{\text{IO}}^{\text{out}}$ as per Equation 5; (vii) $\mathcal{F}_{\text{cutqual}}(\texttt{T}_{\text{IO}}^{\text{out}}, \texttt{C}^{\text{out}}) \in [0,1]$ evaluates visual quality of the shortest path made only of basic actions, similar to $\mathcal{F}_{\text{qual}}$. We set $\alpha_1 = \alpha_2 = \frac{1}{2}$ and $\alpha_3 = \alpha_4 = \alpha_5 = \frac{1}{3}$.

$$
\begin{aligned}
\mathcal{F}_{\text{score}}^{\text{HoCMaze}}(\texttt{T}^{\text{out}}, \texttt{C}^{\text{out}}) =\; &\mathbf{1}\Big[\mathcal{F}_{\text{cov}}(\texttt{T}_{\text{IO}}^{\text{out}}, \texttt{C}^{\text{out}}) = 1, \mathcal{F}_{\text{sol}}(\texttt{T}_{\text{IO}}^{\text{out}}, \texttt{C}^{\text{out}}) = 1, \mathcal{F}_{\text{nocross}}(\texttt{T}_{\text{IO}}^{\text{out}}, \texttt{C}^{\text{out}}) = 1, \\
&\quad \mathcal{F}_{\text{nocut}}(\texttt{T}_{\text{IO}}^{\text{out}}, \texttt{C}^{\text{out}}) = 1, \mathcal{F}_{\text{notred}}(\texttt{T}_{\text{IO}}^{\text{out}}, \texttt{C}^{\text{out}}) = 1\Big] \cdot \\
&\Big[\alpha_1 \mathcal{F}_{\text{cov}}(\texttt{T}_{\text{IO}}^{\text{out}}, \texttt{C}^{\text{out}}) + \alpha_2 \mathcal{F}_{\text{qual}}(\texttt{T}_{\text{IO}}^{\text{out}}, \texttt{C}^{\text{out}})\Big]
\end{aligned}
\tag{3}
$$

$$
\begin{aligned}
\mathcal{F}_{\text{score}}^{\text{Karel}}(\texttt{T}^{\text{out}}, \texttt{C}^{\text{out}}) =\; &\mathbf{1}\Big[\mathcal{F}_{\text{cov}}(\texttt{T}_{\text{IO}}^{\text{out}}, \texttt{C}^{\text{out}}) = 1, \mathcal{F}_{\text{sol}}(\texttt{T}_{\text{IO}}^{\text{out}}, \texttt{C}^{\text{out}}) = 1, \mathcal{F}_{\text{nocross}}(\texttt{T}_{\text{IO}}^{\text{out}}, \texttt{C}^{\text{out}}) = 1, \\
&\quad \mathcal{F}_{\text{nocut}}(\texttt{T}_{\text{IO}}^{\text{out}}, \texttt{C}^{\text{out}}) = 1, \mathcal{F}_{\text{notred}}(\texttt{T}_{\text{IO}}^{\text{out}}, \texttt{C}^{\text{out}}) = 1\Big] \cdot \\
&\Big[\alpha_3 \mathcal{F}_{\text{cov}}(\texttt{T}_{\text{IO}}^{\text{out}}, \texttt{C}^{\text{out}}) + \alpha_4 \mathcal{F}_{\text{qual}}(\texttt{T}_{\text{IO}}^{\text{out}}, \texttt{C}^{\text{out}}) + \alpha_5 \mathcal{F}_{\text{cutqual}}(\texttt{T}_{\text{IO}}^{\text{out}}, \texttt{C}^{\text{out}})\Big]
\end{aligned}
\tag{4}
$$

We use the same measure of visual quality for both domains, keeping into account the number of moves, turns, segments, long-segments, and turn-segments, as explained next. More specifically, segments and long-segments correspond to consecutive sequences of 'moves' containing more than 3 and 5 actions, respectively; turn-segments correspond to consecutive sequences of 'turnLeft' or 'turnRight' containing more than 3 actions. The formula for $\mathcal{F}_{\text{qual}}$ is given in Equation 5 below; we also clip the values for each counter # w.r.t. its corresponding normalization factor (not depicted here for brevity).

$$
\begin{aligned}
\mathcal{F}_{\text{qual}}(\texttt{T}^{\text{out}}, \texttt{C}^{\text{out}}) =\; &\frac{3}{4} \cdot \left(\frac{1}{4} \cdot \left(\frac{\#\text{moves}}{2n} + \frac{\#\text{turns}}{n} + \frac{\#\text{segments}}{n/2} + \frac{\#\text{long-segments}}{n/3}\right)\right) + \\
&\frac{1}{4} \cdot \left(1 - \frac{\#\text{turn-segments}}{n/2}\right)
\end{aligned}
\tag{5}
$$

## E.2  Synthetic Task Specifications

**Domain-specific elements.**  As introduced in Sections 2 and 4, we use two DSLs shown in Figures 17a and 17b, adapted from the DSLs in (Bunel et al., 2018; Ahmed et al., 2020).

```
code C     := def RUN () DO y
rule y     := s | g | s; g
rule s     := a | s; s | IF (b) DO s
              | IF (b) DO s ELSE s
              | REPEAT (x) DO s
rule g     := REPEATUNTIL (goal) DO s
action a   := move | turnLeft | turnRight
bool b     := pathAhead | pathLeft | pathRight
iter x     := 2 | 3 | 4 | 5 | 6 | 7 | 8 | 9 | 10
```

(a) DSL for HoCMaze domain

```
code C     := def RUN () DO s
rule s     := a | s; s | IF (b) DO s | IF (b) DO s ELSE s
              | WHILE (b) DO s | REPEAT (x) DO s
action a   := move | turnLeft | turnRight
              | putMarker | pickMarker
bool b     := pathAhead | pathLeft | pathRight
              | no-pathAhead | markerPresent | no-markerPresent
iter x     := 2 | 3 | 4 | 5 | 6 | 7 | 8 | 9 | 10
```

(b) DSL for Karel domain

| Domain | All | Easy | | | Hard | |
|---|---|---|---|---|---|---|
| | | (depth, constructs) | | | (depth, constructs) | |
| | | $(1, 0)$ | $(2, 1)$ | $(2, 2)$ | $(3, 2)$ | $(3, 3)$ |
| HoCMaze | $1,016$ | $183$ | $69$ | $47$ | $136$ | $581$ |
| Karel | $1,027$ | $300$ | $155$ | $277$ | $295$ | $0$ |

(c) Dataset of synthetic task specifications

Figure 17: **(a)** Synthetic datasets used for training/evaluation in Section 4). **(b)** DSLs for two domains.

---

**Algorithm 1:** Specification Dataset Collection Procedure

---

**Input:** list $\mathbb{S}$ of tuples (code structure $s$, required size $l$); maximum candidate set size $m$;

$\mathbb{D} \leftarrow \emptyset$ ;                                    /* Dataset initialized to empty set */

**foreach** $(s, l) \in \mathbb{S}$ **do**

  $\mathbb{C} \leftarrow \emptyset$ ;                              /* Candidate set initialized to empty set */

  **while** size($\mathbb{C}$) $< m$ **do**

    code $\leftarrow$ GenerateCode($s$);

    task $\leftarrow$ TASKORACLE(code);

    score $\leftarrow \mathcal{F}_{\text{score}}$(task, code);

    **if** score $> 0$ **then**

      add (code, task, score) to $\mathbb{C}$;

  sort $\mathbb{C}$ according to the score, in decreasing order;

  counter $\leftarrow 0$;

  **while** counter $< l$ and $\mathbb{C} \neq \emptyset$ **do**

    (code, task, score) $\leftarrow$ Pop($\mathbb{C}$);

    accept $\leftarrow$ Inspect(task, code) ;                    /* Inspection step */

    **if** accept **then**

      $\psi \leftarrow$ ExtractSpecs(code);

      add $\psi$ to $\mathbb{D}$;

      counter $\leftarrow$ counter+1;

**Output:** Dataset $\mathbb{D}$;

---

**Dataset.** We follow Algorithm 1 to create dataset $\mathbb{D}$. For each code structure, we generate a set of candidate codes and obtain an oracle task for these codes. We filter them out if a low-quality task is obtained, supplementing this filtering with an additional inspection step. This inspection step is necessary because semantic irregularities (e.g., IFELSE with the same IF and ELSE bodies) can get past the previous filtering step. As the compute and implementation efforts are larger for an automatic system that would detect such irregularities, which are easy to spot, we opt for a direct inspection step. Figure 17c provides a summary of datasets $\mathbb{D}$ for each domain.

### E.3 Training Process

**Training the code generator.** We employ a standard approach, using an imitation (supervised) form of learning, with a cross-entropy loss for an LSTM-based architecture (see Appendix D) (Devlin et al., 2017; Bunel et al., 2018). We augment $\mathbb{D}^{\text{train}}$ by adding all the possible combinations of construct instantiations for a given code. The training plots and the hyperparameters used can be seen in Figure 18. We report the validation performance smoothed via an exponential decay function, and the batch loss averaged over one epoch.

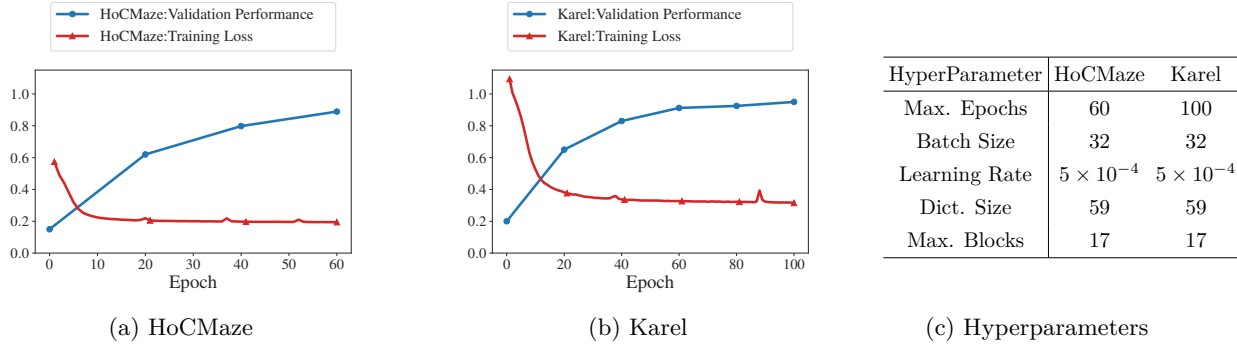

|                | (a) HoCMaze | (b) Karel | (c) Hyperparameters |

Figure 18: Illustration of training details for the code generator. (a) and (b) show the training curves with mean epoch loss and validation performance, based on metric $\mathcal{M}$, for both the HoCMaze and Karel domains. (c) shows the hyperparameters employed for the code generator training.

**Training the puzzle generator.** We use an RL procedure, using the instantiations of $\mathcal{F}_{\text{score}}$ as rewards. We augment the RL training set with additional codes produced by the previously trained code generator. To encourage higher quality tasks, we use a form of curriculum as follows: after a certain epoch, we give a reward larger than 0 only if the ratio between the scores of the output task and the TASKORACLE's task is larger than a factor $\widehat{\lambda}_2$; we gradually increase $\widehat{\lambda}_2$ from 0.8 to 0.9. For Karel, we also employ a temperature parameter during training, encouraging exploration during inference. The training plots and the hyperparameters used can be seen in Figures 19. We report the validation performance (i.e., same metric employed for NEURPUZZLEGEN) smoothed via an exponential decay function, and the batch reward averaged over one epoch.

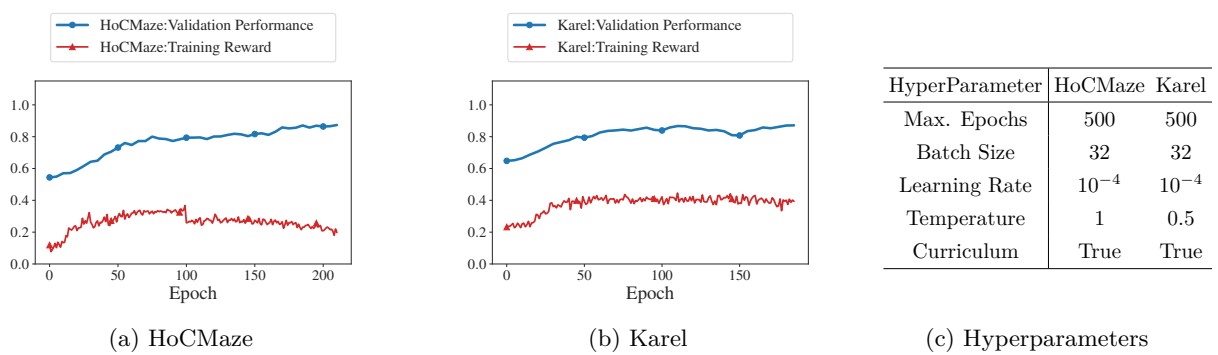

|                | (a) HoCMaze | (b) Karel | (c) Hyperparameters |

Figure 19: Illustration of training details for the puzzle generator. (a) and (b) show the training curves with mean epoch reward and validation performance, based on metric $\mathcal{M}$ for both the HoCMaze and Karel domains. A form of curriculum learning was employed, which explains the lack of general monotonicity for the reward. (c) shows the hyperparameters employed for the puzzle generator training.

**Further implementation details.** We limit the number of possible initial locations for a grid to one representative per quadrant. In total, we consider 5 quadrants (i.e., top-left, bottom-left, center, top-right, bottom-right). We do this to limit the action space to a more tractable amount for a variable grid size. With 5 quadrants and 4 possible orientations, this leads to $5 \times 4 = 20$ possible initial location/orientation pairs, offering already enough variability.

**Detailed results on synthetic task specifications.** Figure 20 reports evaluation results for different techniques for a fixed number of code/puzzle rollouts across two domains and segments, complementary to Figure 5.

| Technique | HoCMaze | | | Karel | | |
|---|---|---|---|---|---|---|
| | All | Easy | Hard | All | Easy | Hard |
| BASETASKSYN$_{c:5,p:10}$ | 13.6 (1.3) | 46.2 (4.1) | 2.0 (1.3) | 35.7 (1.0) | 49.3 (0.9) | 10.9 (2.7) |
| NEURTASKSYN$_{c:5,p:10}$ | 81.4 (3.7) | 100.0 (0.0) | 74.3 (5.1) | 92.6 (1.4) | 100.0 (0.0) | 79.3 (3.9) |
| BASEPUZZLEGEN$_{c:FIX,p:10}$ | 55.6 (1.8) | 91.7 (2.4) | 41.9 (2.3) | 71.8 (3.8) | 86.6 (3.8) | 45.0 (3.9) |
| NEURPUZZLEGEN$_{c:FIX,p:10}$ | 78.4 (2.5) | 100.0 (0.0) | 70.3 (3.4) | 79.8 (0.6) | 92.0 (1.3) | 57.7 (1.8) |

Figure 20: Results on synthetic task specifications for HoCMaze and Karel; see Figure 17c and Section 4.

# F    Experimental Evaluation with Real-World Task Specifications: Details

## F.1    Real-World Task Specifications

In Figures 21 and 22 below, we list source tasks T and codes C for 10 task specifications mentioned in Figure 6.

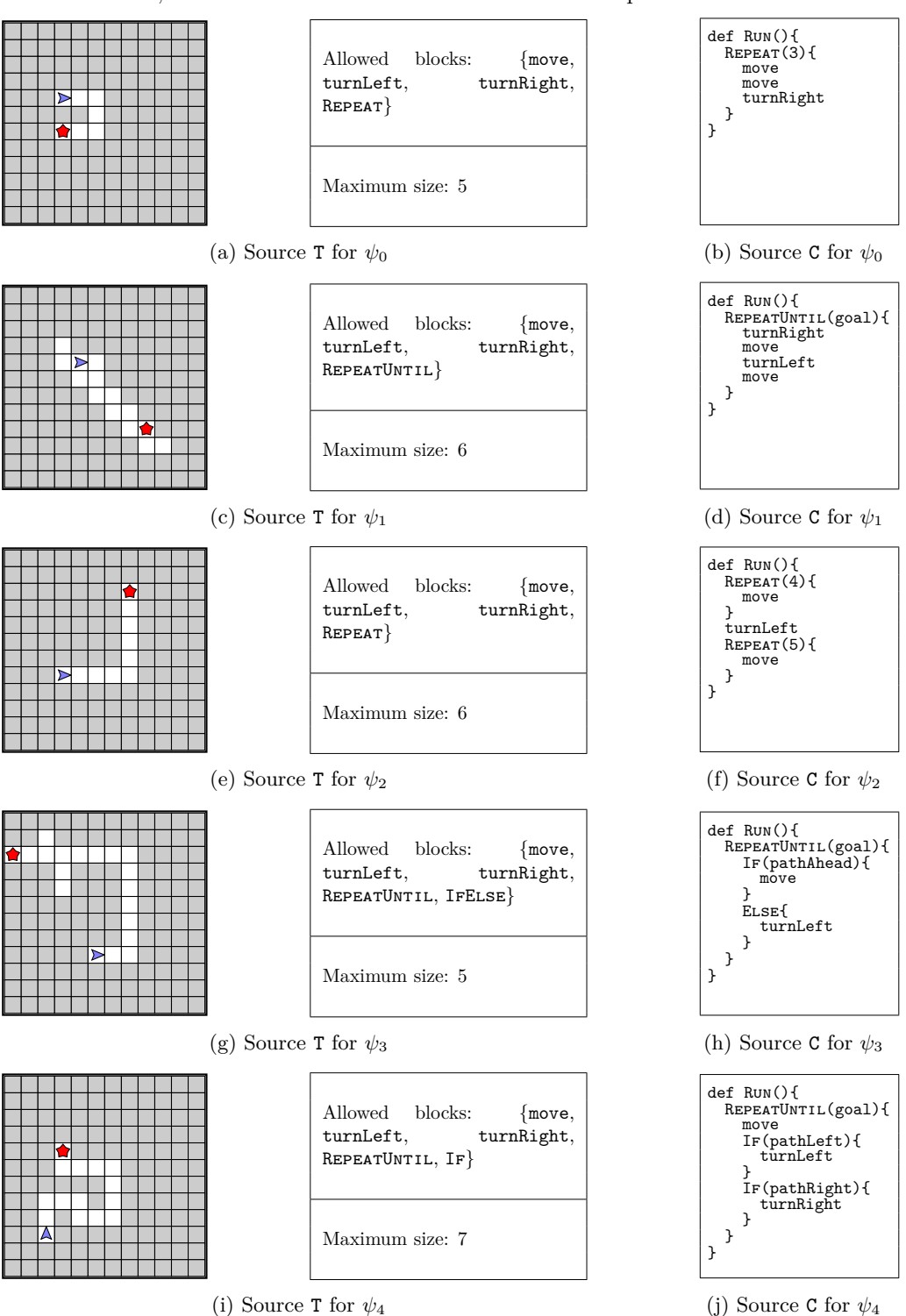

Figure 21: Overview of HoCMaze sources.

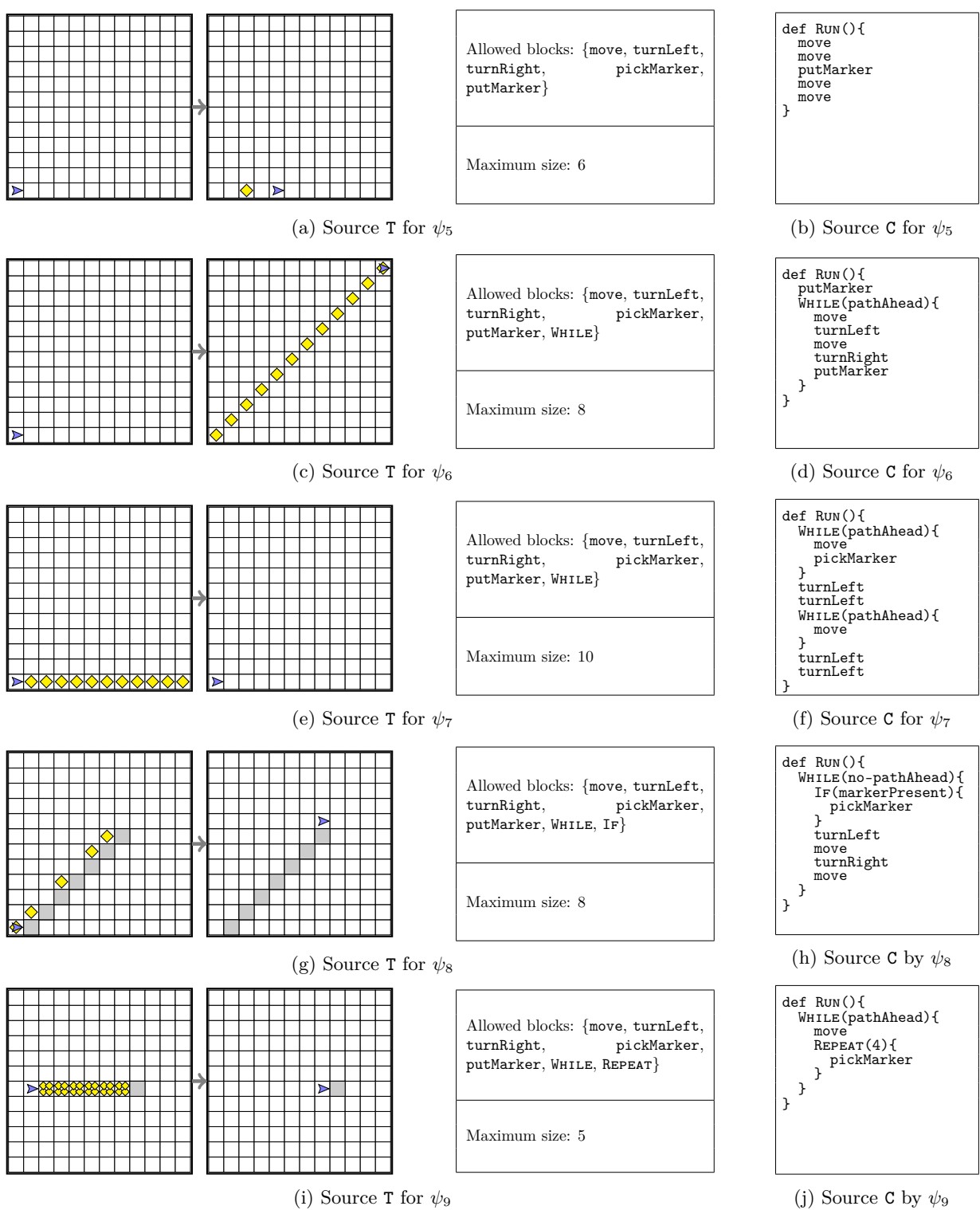

Figure 22: Overview of Karel sources.

## F.2   GPT4TaskSyn

We describe next the details of our interaction with GPT-4 for generating the visual puzzles $\texttt{T}_{\text{IO}}^{\text{out}}$ and the intermediate code $\texttt{C}^{\text{out}}$ via the GPT4TASKSYN techniques. Our interaction is conducted through the platform (OpenAI, 2023a). We try several strategies and prompts to make GPT-4 work for synthesizing visual programming tasks, as it tends to struggle with logical and spatial reasoning. Thus, we opt for a two-stage task synthesis process which works the best. We first ask GPT-4 to generate a code $\texttt{C}^{\text{out}}$ for $\psi^{\text{in}}$, by using 5 separate queries.

For the GPT4TASKSYN-*converse* technique, we start with an initial prompt and then use follow-up prompts to fix any mistakes, as GPT-4 occasionally ignores part of the specifications. The initial and follow-up prompts used for generating $\texttt{C}^{\text{out}}$ are presented in Figures 23a and 24a. We select the best code generated during the 5 separate queries based on our expertise. The second stage comprises of additional 5 separate queries for generating $\texttt{T}_{\text{IO}}^{\text{out}}$ for the selected code $\texttt{C}^{\text{out}}$. Again, we start with an initial prompt and then use follow-up prompts to fix any issues. The follow-up prompts are necessary because GPT-4 tends to struggle with spatial orientation and with the relationship between $\texttt{C}^{\text{out}}$ and $\texttt{T}_{\text{IO}}^{\text{out}}$. The initial and follow-up prompts used for generating $\texttt{T}_{\text{IO}}^{\text{out}}$ are presented in Figures 23b and 24b. Similar to the code selection process, we select the best visual puzzle generated during the 5 separate queries based on our expertise. Once we get $\texttt{C}^{\text{out}}$ and $\texttt{T}_{\text{IO}}^{\text{out}}$, we set $\texttt{T}_{\text{code}}^{\text{out}}$ as for BASETASKSYN and NEURTASKSYN.

For the GPT4TASKSYN-*fewshot*, technique, we employ a few-shot approach for both stages, by first giving 3 synthesis examples (i.e., for code and task synthesis, respectively). We do not use follow-up prompts here. The few-shot prompts are presented in Figures 25 and 26.

---

**Code: Initial prompt**

I am working in the block-based visual programming domain of Hour of Code: Maze Challenge from code.org. In this domain, the following types of coding blocks are available:

- Basic action blocks: move forward, turn left, turn right.

- Boolean conditions: path ahead, path left, path right.

- Loops: repeatUntil(goal), repeat(int).

- Conditionals: if(boolean), if(boolean)else.

In this domain, a task is represented as an 8x8 visual grid that contains WALL cells, FREE cells, AVATAR (with specific location and direction), and GOAL. We represent a task's 8x8 visual grid with the following symbols.

# represents a WALL cell.

+ represents a FREE cell.

* represents GOAL.

E represents AVATAR's location facing East direction.

W represents AVATAR's location facing West direction.

N represents AVATAR's location facing North direction.

S represents AVATAR's location facing South direction.

Below, I am giving you a program structure. Can you generate a code that respects this program structure?

— Structure —

[SKETCH]

You should not change the structure. This means that you shouldn't add or remove any loops (e.g., repeatUntil(goal), repeat(int)) and conditionals (e.g., if(boolean), if(boolean)else). The program needs to be valid, meaning that bodies of constructs cannot remain empty. To complete this given structure, you can use basic action blocks, boolean conditions, and iteration numbers that are available in the Hour of Code: Maze Challenge programming.

— Code —

---

**Code: Follow-up prompt in case of constructs changed**

Your code does not follow the program structure I have given. You shouldn't add or remove any loops (e.g., repeatUntil(goal), repeat(int)) and conditionals (e.g., if(boolean), if(boolean)else). Can you try to generate a new code for the same structure?

---

**Code: Follow-up prompt for any other issues**

Your code could be improved! You can think of producing a better code by reasoning about the AVATAR's actions when the code is executed. Can you try to generate a new code respecting the program structure I have given?

(a) Prompts used for obtaining $\mathtt{C}^{\mathrm{out}}$

---

**Task: Initial prompt**

I am working in the block-based visual programming domain of Hour of Code: Maze Challenge from code.org. In this domain, the following types of coding blocks are available:

- Basic action blocks: move forward, turn left, turn right.

- Boolean conditions: path ahead, path left, path right.

- Loops: repeatUntil(goal), repeat(int).

- Conditionals: if(boolean), if(boolean)else.

In this domain, a task is represented as an 8x8 visual grid that contains WALL cells, FREE cells, AVATAR (with specific location and direction), and GOAL. We represent a task's 8x8 visual grid with the following symbols.

\# represents a WALL cell.

+ represents a FREE cell.

\* represents GOAL.

E represents AVATAR's location facing East direction.

W represents AVATAR's location facing West direction.

N represents AVATAR's location facing North direction.

S represents AVATAR's location facing South direction.

Below I am giving you a solution code. Can you generate a task with 8x8 visual grid that would be solved by this code?

— Solution —

[CODE]

The visual grid must contain AVATAR (with specific location and direction) along with GOAL, and can have WALL cells and FREE cells. Number your grid with row numbers (1 to 8) and column numbers (1 to 8). Also, you should tell me the position of AVATAR and GOAL in your generated task so we are sure about the numbering.

You can verify the correctness of your generated task by executing the solution code on your task. A solution code for a task takes AVATAR to GOAL when executed. Note that AVATAR can only move on FREE cells and will crash if it tries to go to a WALL cell. If your generated task is not correct, you should try again to generate a correct task.

— Task —

---

**Task: Follow-up prompt for any issues**

Your code does not solve the generated grid. Be careful with the AVATAR as it should reach the goal after the code execution. Keep the code fixed. Can you try to generate a new visual grid and explain your reasoning? Recall that your code, when executed, should take the AVATAR from its initial location to the GOAL.

---

(b) Prompts used for obtaining the main part of $\text{T}^{\text{out}}_{\text{IO}}$

Figure 23: Prompts used in the implementation of GPT4TaskSyn-*converse* technique for HoCMaze domain.

---

**Code: Initial prompt**

I am working in the block-based visual programming domain of Karel programming. In this domain, the following types of coding blocks are available:

- Basic action blocks: move forward, turn left, turn right, pick marker, put marker.

- Boolean conditions: path ahead, path left, path right, marker present, no path ahead, no marker present.

- Loops: while(boolean), repeat(int).

- Conditionals: if(boolean), if(boolean)else.

In this domain, a task is represented as a pair of 10x10 visual pregrid and 10x10 visual postgrid. This pregrid and postgrid contain WALL cells, FREE cells, AVATAR (with specific location and direction), and markers. We represent a task's 10x10 visual pregrid and postgrid with the following symbols.

\# represents a WALL cell.

+ represents a FREE cell.

m represents a cell with marker.

E represents AVATAR's location on a cell without marker, facing East direction.

W represents AVATAR's location on a cell without marker, facing West direction.

N represents AVATAR's location on a cell without marker, facing North direction.

S represents AVATAR's location on a cell without marker, facing South direction.

Em represents AVATAR's location on a cell with marker, facing East direction.

Wm represents AVATAR's location on a cell with marker, facing West direction.

Nm represents AVATAR's location on a cell with marker, facing North direction.

Sm represents AVATAR's location on a cell with marker, facing South direction.

Below, I am giving you a program structure. Can you generate a code that respects this program structure?

—- Structure —-

[SKETCH]

You should not change the structure. This means that you shouldn't add or remove any loops (e.g., while(boolean), repeat(int)) and conditionals (e.g., if(boolean), if(boolean)else). The program needs to be valid, meaning that bodies of constructs cannot remain empty. To complete this given structure, you can use basic action blocks, boolean conditions, and iteration numbers that are available in Karel programming.

— Code —

---

**Code: Follow-up prompt in case of constructs changed**

Your code does not follow the programming structure I have given. You shouldn't add or remove any loops (e.g., while(boolean), repeat(int)) and conditionals (e.g., if(boolean), if(boolean)else). Can you try to generate a new code for the same structure?

---

**Code: Follow-up prompt for any other issues**

Your code could be improved! You can think of producing a better code by reasoning about the Karel AVATAR when the code is executed. Can you try to generate a new code?

---

(a) Prompts used for obtaining $C^{out}$

---

**Task: Initial prompt**

I am working in the block-based visual programming domain of Karel programming. In this domain, the following types of coding blocks are available:

- Basic action blocks: move forward, turn left, turn right, pick marker, put marker.

- Boolean conditions: path ahead, path left, path right, marker present, no path ahead, no marker present.

- Loops: while(boolean), repeat(int).

- Conditionals: if(boolean), if(boolean)else.

In this domain, a task is represented as a pair of 10x10 visual pregrid and 10x10 visual postgrid. This pregrid and postgrid contain WALL cells, FREE cells, AVATAR (with specific location and direction), and markers. We represent a task's 10x10 visual pregrid and postgrid with the following symbols.

# represents a WALL cell.

+ represents a FREE cell.

m represents a cell with marker.

E represents AVATAR's location on a cell without marker, facing East direction.

W represents AVATAR's location on a cell without marker, facing West direction.

N represents AVATAR's location on a cell without marker, facing North direction.

S represents AVATAR's location on a cell without marker, facing South direction.

Em represents AVATAR's location on a cell with marker, facing East direction.

Wm represents AVATAR's location on a cell with marker, facing West direction.

Nm represents AVATAR's location on a cell with marker, facing North direction.

Sm represents AVATAR's location on a cell with marker, facing South direction.

Below I am giving you a solution code. Can you generate a task with a pair of 10x10 visual pregrid and 10x10 visual postgrid that would be solved by this code?

— Solution —

[CODE]

Both the visual pregrid and visual postgrid must contain AVATAR (with specific location and direction), and can have WALL cells, FREE cells, and markers. Number your grids with row numbers (1 to 10) and column numbers (1 to 10). Also, you should tell me the position of AVATAR in your generated pregrid and postgrid so we are sure about the numbering.

You can verify the correctness of your generated task by executing the solution code on your task. A solution code for a task transforms the pregrid into the postgrid when executed. Note that AVATAR can only move on FREE cells and will crash if it tries to go to a WALL cell. If your generated task is not correct, you should try again to generate a correct task.

— Task —

---

**Task: Follow-up prompt for any issues**

Your code does not solve the generated pregrid and postgrid. Be careful with the AVATAR in the postgrid as it should show the effect of the code execution. Keep the code fixed. Can you try to generate a new visual pregrid and postgrid and explain your reasoning? Recall that your code, when executed, should transform the pregrid into the postgrid. Be careful with the AVATAR in the postgrid as it should show the effect of the code execution.

---

(b) Prompts used for obtaining the main part of $\mathrm{T_{IO}^{out}}$

Figure 24: Prompts used in the implementation of GPT4TASKSYN-*converse* technique for Karel domain.

---

**Code: Few-shot prompt**

I am working in the block-based visual programming domain of Hour of Code: Maze Challenge from code.org. In this domain, the following types of coding blocks are available:

- Basic action blocks: move forward, turn left, turn right.

- Boolean conditions: path ahead, path left, path right.

- Loops: repeatUntil(goal), repeat(int).

- Conditionals: if(boolean), if(boolean)else.

In this domain, a task is represented as an 8x8 visual grid that contains WALL cells, FREE cells, AVATAR (with specific location and direction), and GOAL. We represent a task's 8x8 visual grid with the following symbols.

# represents a WALL cell.

+ represents a FREE cell.

* represents GOAL.

E represents AVATAR's location facing East direction.

W represents AVATAR's location facing West direction.

N represents AVATAR's location facing North direction.

S represents AVATAR's location facing South direction.

Below, I will give you a program structure. Can you generate a code that respects this program structure?

You should not change the structure. This means that you shouldn't add or remove any loops (e.g., repeatUntil(goal), repeat(int)) and conditionals (e.g., if(boolean), if(boolean)else). The program needs to be valid, meaning that bodies of constructs cannot remain empty. To complete this given structure, you can use basic action blocks, boolean conditions, and iteration numbers that are available in the Hour of Code: Maze Challenge programming.

I am giving you some examples comprising a program structure and a code that respects the structure. Provide the code for the last structure.

— Example i: Structure —

[SKETCH_i]

— Example i: Code —

[CODE_i]

— Example n: Structure —

[SKETCH_n]

— Example n: Code —

(a) Prompts used for obtaining $\mathtt{C}^{\mathrm{out}}$

---

**Task: Few-shot prompt**

I am working in the block-based visual programming domain of Hour of Code: Maze Challenge from code.org. In this domain, the following types of coding blocks are available.

- Basic action blocks: move forward, turn left, turn right.

- Boolean conditions: path ahead, path left, path right.

- Loops: repeatUntil(goal), repeat(int).

- Conditionals: if(boolean), if(boolean)else.

In this domain, a task is represented as an 8x8 visual grid that contains WALL cells, FREE cells, AVATAR (with specific location and direction), and GOAL. We represent a task's 8x8 visual grid with the following symbols.

\# represents a WALL cell.

\+ represents a FREE cell.

\* represents GOAL.

E represents AVATAR's location facing East direction.

W represents AVATAR's location facing West direction.

N represents AVATAR's location facing North direction.

S represents AVATAR's location facing South direction.

Below I will give you a solution code. Can you generate a task with 8x8 visual grid that would be solved by this code?

The visual grid must contain AVATAR (with specific location and direction) along with GOAL, and can have WALL cells and FREE cells. Number your grid with row numbers (1 to 8) and column numbers (1 to 8). Also, you should tell me the position of AVATAR and GOAL in your generated task so we are sure about the numbering.

You can verify the correctness of your generated task by executing the solution code on your task. A solution code for a task takes AVATAR to GOAL when executed. Note that AVATAR can only move on FREE cells and will crash if it tries to go to a WALL cell. If your generated task is not correct, you should try again to generate a correct task.

I am giving you some examples comprising a solution code and task that is solved by this code. Provide the task for the last solution code.

— Example i: Solution —

[CODE_i]

— Example i: Task —

[TASK_i]

— Example n: Solution —

[CODE_n]

— Example n: Task —

(b) Prompts used for obtaining the main part of $\mathrm{T_{IO}^{out}}$

Figure 25: Prompts used in the implementation of GPT4TASKSYN-*fewshot* technique for HoCMaze domain.

---

**Code: Few-shot prompt**

I am working in the block-based visual programming domain of Karel programming. In this domain, the following types of coding blocks are available:

- Basic action blocks: move forward, turn left, turn right, pick marker, put marker.

- Boolean conditions: path ahead, path left, path right, marker present, no path ahead, no marker present.

- Loops: while(boolean), repeat(int).

- Conditionals: if(boolean), if(boolean)else.

In this domain, a task is represented as a pair of 10x10 visual pregrid and 10x10 visual postgrid. This pregrid and postgrid contain WALL cells, FREE cells, AVATAR (with specific location and direction), and markers. We represent a task's 10x10 visual pregrid and postgrid with the following symbols.

\# represents a WALL cell.

+ represents a FREE cell.

m represents a cell with marker.

E represents AVATAR's location on a cell without marker, facing East direction.

W represents AVATAR's location on a cell without marker, facing West direction.

N represents AVATAR's location on a cell without marker, facing North direction.

S represents AVATAR's location on a cell without marker, facing South direction.

Em represents AVATAR's location on a cell with marker, facing East direction.

Wm represents AVATAR's location on a cell with marker, facing West direction.

Nm represents AVATAR's location on a cell with marker, facing North direction.

Sm represents AVATAR's location on a cell with marker, facing South direction.

Below, I will give you a program structure. Can you generate a code that respects this program structure?

You should not change the structure. This means that you shouldn't add or remove any loops (e.g., while(boolean), repeat(int)) and conditionals (e.g., if(boolean), if(boolean)else). The program needs to be valid, meaning that bodies of constructs cannot remain empty. To complete this given structure, you can use basic action blocks, boolean conditions, and iteration numbers that are available in Karel programming.

I am giving you some examples comprising a program structure and a code that respects the structure. Provide the code for the last structure.

— Example i: Structure —

[SKETCH_i]

— Example i: Code —

[CODE_i]

— Example n: Structure —

[SKETCH_n]

— Example n: Code —

(a) Prompts used for obtaining $\mathtt{C}^{\mathrm{out}}$

---

**Task: Few-shot prompt**

I am working in the block-based visual programming domain of Karel programming. In this domain, the following types of coding blocks are available.

- Basic action blocks: move forward, turn left, turn right, pick marker, put marker.

- Boolean conditions: path ahead, path left, path right, marker present, no path ahead, no marker present.

- Loops: while(boolean), repeat(int).

- Conditionals: if(boolean), if(boolean)else.

In this domain, a task is represented as a pair of 10x10 visual pregrid and 10x10 visual postgrid. This pregrid and postgrid contain WALL cells, FREE cells, AVATAR (with specific location and direction), and markers. We represent a task's 10x10 visual pregrid and postgrid with the following symbols.

\# represents a WALL cell.

\+ represents a FREE cell.

m represents a cell with marker.

E represents AVATAR's location on a cell without marker, facing East direction.

W represents AVATAR's location on a cell without marker, facing West direction.

N represents AVATAR's location on a cell without marker, facing North direction.

S represents AVATAR's location on a cell without marker, facing South direction.

Em represents AVATAR's location on a cell with marker, facing East direction.

Wm represents AVATAR's location on a cell with marker, facing West direction.

Nm represents AVATAR's location on a cell with marker, facing North direction.

Sm represents AVATAR's location on a cell with marker, facing South direction.

Below I will give you a solution code. Can you generate a task with a pair of 10x10 visual pregrid and 10x10 visual postgrid that would be solved by this code?

Both the visual pregrid and visual postgrid must contain AVATAR (with specific location and direction), and can have WALL cells, FREE cells, and markers. Number your grids with row numbers (1 to 10) and column numbers (1 to 10). Also, you should tell me the position of AVATAR in your generated pregrid and postgrid so we are sure about the numbering.

You can verify the correctness of your generated task by executing the solution code on your task. A solution code for a task transforms the pregrid into the postgrid when executed. Note that AVATAR can only move on FREE cells and will crash if it tries to go to a WALL cell. If your generated task is not correct, you should try again to generate a correct task.

I am giving you some examples comprising a solution code and task that is solved by this code. Provide the task for the last solution code.

— Example i: Solution —

[CODE_i]

— Example i: Task Pregrid —

[TASK_PREGRID_i]

– Example i: Task Postgrid —

[TASK_POSTGRID_i]

— Example n: Solution —

[CODE_n]

— Example n: Task Pregrid —

---

(b) Prompts used for obtaining the main part of $T_{IO}^{out}$

Figure 26: Prompts used in the implementation of GPT4TASKSYN-*fewshot* technique for Karel domain.

