# OpenReview forum: "Neural Task Synthesis for Visual Programming"
_TMLR — Accepted by TMLR_

### Review · Reviewer_hZEs · 2023-10-17

**Summary Of Contributions:**

The authors study creation of introductory programming content, specifically based on drag-and-drop block style programming (i.e. like Scratch). The goal is to automatically create good programming puzzles, scored according to various criteria (an optimal solution of short length, solutions require demonstrating correct usage of programming concepts, etc.) These criteria are summarized into a final "score" for the quality of a puzzle, which can be treated as a scoring signal for an RL type method.

Separately from this, the authors study the combination of learned neural-net systems with symbolic techniques. We assume access to a baseline symbolic engine for puzzle generation, where state is the current puzzle + code execution, and the engine is driven by actions deciding how to modify the puzzle (i.e. that a certain branch of the if-block should be taken and the puzzle should support this.). One baseline runs this engine using random search whenever an action would be taken. Another baseline runs GPT-4 directly with a prompt to generate a code solution + a corresponding puzzle.

The method proposed by the authors is to

1) Generate a dataset of synthetic puzzle examples by sampling code + initial constraints on programming constructs
2) Train a policy (LSTM-based) with RL based on our scoring of what a good puzzle looks like, where the policy's actions drive the symbolic engine.

The resulting method is compared on some existing introductory programming environments (i.e. Hour of Code)

**Audience:**

Yes

**Broader Impact Concerns:**

No concerns here.

**Claims And Evidence:**

Yes

**Requested Changes:**

For making the paper clearer, my recommendation would be to modify Figure 1a to describe, in natural language, what the goal of the task synthesis specification is. Just saying "the DSL constraints specify that boolean conditions should be in this set and use <= 7 blocks" doesn't  help provide context on those constraints are for. Just a sentence describing the high-level goal would be helpful. Something like "Generate a grid with start and end such that a coding solution satisfies (these constraints) and matches (this block structure)"

**Strengths And Weaknesses:**

I found it tricky to follow the paper. Perhaps that's because I am less familiar with task synthesis literature, but in general I found the description of the problem statement and solution to be too abstract until I got further into experiments and played around with Hour of Code / Code.org myself to understand what was going on.

Once I understood what the paper was doing, I thought it was good. However I do wonder about a "GPT-4 driven symbolic engine" baseline. My understanding is that GPT-4 is asked to generate the puzzle without using the symbolic engine in any way. There are a few papers that examine LLMs + tool use, i.e. the prompt describes how the model could use a browser, calculator, etc., in the style of Code-as-Policies / Language Model Programs (https://code-as-policies.github.io/). And I was curious at what would happen if a prompted GPT-4 was used in place of the Neural Model in Figure 4b, i.e. the prompt would be "here is the state of a helpful puzzle making tool, here are the actions you could take, please output exactly one", or something along those lines. Essentially I am wondering how necessary the RL step is here.

---

> ### Author Response · Authors · 2023-11-23
> **Response to Reviewer hZEs**
>
> Thank you for carefully reviewing our paper! We greatly appreciate your feedback. Please see below our responses to your comments.
>
> -----
>
> **1. in general I found the description of the problem statement and solution to be too abstract…**
>
> In the revised version of the paper, we have updated Sections 2, 3, and 4. Please refer to the common response for more details about those changes.
>
> -----
>
> **2. I do wonder about a "GPT-4 driven symbolic engine" baseline … I am wondering how necessary the RL step is here.**
>
> We thank the reviewer for bringing this point up for discussion. We believe that using GPT-4 to guide every decision may be too costly. As shown in Figure 3, synthesizing one puzzle requires multiple sequential decisions, each involving an API query to GPT-4. The higher-quality puzzles typically require a few dozen queries per task for one puzzle rollout. Furthermore, we would need multiple rollouts per task to obtain a high-quality task (as mentioned in Sections 4 and 5), leading to hundreds of API queries to GPT-4.
>
> Nevertheless, we appreciate the reviewer’s suggestion and idea. An alternate approach would be to use a smaller but open-source model (e.g., Llama-2). We expect that some fine-tuning would be needed so that the model can guide the symbolic engine to reach high-quality tasks. This is indeed an exciting direction for future work.
>
> -----
>
> **3. To make the paper clearer, my recommendation would be to modify Figure 1a to describe, in natural language, what the goal of the task synthesis specification is…**
>
> We thank the reviewer for this suggestion. We have substituted Figure 1 with an illustration that gives an overview of an AI system that provides support to a student struggling to solve a given task. The AI system helps the student by generating new examples or simple tasks. We believe that this adjustment better motivates our work.
>
>
> -----
>
> We hope that our responses and the revised version of the paper can address your concerns and are helpful in improving your rating. If you have any other comments or feedback, please let us know! We are looking forward to hearing back from you! Thank you again for the review.

---

### Review · Reviewer_Lkyo · 2023-10-27

**Summary Of Contributions:**

Broadly, this work is interested in generating new visual programming exercises consisting of small 2D mazes, code that solves the maze, and some restrictions in allowed code commands and allowed code structure.
The work contributes:
- (1) a novel method for generating these that's faster and more robust than existing methods.
- (2) a synthetic dataset for training the method
- (3) evaluation of the method on self-defined metrics

**Audience:**

No

**Broader Impact Concerns:**

No concerns.

**Claims And Evidence:**

No

**Requested Changes:**

- **R.1** I would require a rewrite of the sections 2, 3, and 4 for clarity and some additional figures to illustrate how everything works together and what data exactly is passed between the different subsystems.
- **R.2** I need a convincing argument (a) that this is faster than existing methods (to back up contribution (1), and (b) that this is beneficial to the wider TMLR audience.

My current assessment is:

**Are the claims made in the submission supported by**
- accurate, -> possibly?
- convincing and -> no, I cannot follow
- clear evidence -> no

**Would at least some individuals in TMLR's audience be interested in knowing the findings of this paper?**
- I honestly don't know who this would apply to or who would benefit from this.

**Strengths And Weaknesses:**

### Strengths

- **S.1** I appreciate that the authors included code and task specifications with their submission.

### Weaknesses

- **W.1** The work is really hard to parse. And this is also my main reason for rejection - I couldn't reproduce this because I don't understand what's going on. In the main body of the paper, you're hinting at different methods that are only explained in the appendix but that are crucial to understanding the paper. Only on pages 25 and 26, deep in the appendix was I able to find some crucial details of the algorithm. Other points that fall under this:
    - **W.1.1** "BaseTaskSyn" is first mentioned on page 2 but only explained on page 8
    - **W.1.2** I'm not a computer, so it takes a fair bit of my working memory to execute the code from Fig. 1 and Fig. 2 and apply that to the corresponding maze. You couldn't include a brief summary in the figure caption, what the differences are between the methods? Or maybe include a frame-by-frame "animation" of rollouts and how they fail in the respective cases.
    - **W.1.3** I'm not sure why you're using imitation learning for part of your method and reinforcement learning for the other part of your method
    - **W.1.4** In general, I have no idea what data comes from different systems, how they're trained, and how everything comes together. Your Fig. 3 is a first step towards this but I'd love if you could also show how exactly the code generator works, i.e. what format the incoming and outgoing data has and how it's trained (and how you're delineating the symbolic and neural parts). Same for the puzzle generator. Again, there was an attempt made here in Fig. 4 but I don't know what "state (puzzle/code)" actually means. How is that represented? And what exactly do the decision logits correspond to and change in the symbolic engine? They move the agent, they spawn objects, they remove wall, create walls? How does this guarantee the right number of steps?
    - **W.1.5** There's a mention of a "TaskOracle" that "does one million symbolic executions of a given code C returns highest-scoring task w.r.t. scoring function" - I have no idea what any of this means.
- **W.2** I don't know who this is relevant to. Why is it important for anyone to generate these mazes fast? When does it ever happen that you cannot pre-generate a couple offline and sample them? Also, what is the actual speed difference?

---

> ### Author Response · Authors · 2023-11-23
> **Response to Reviewer Lkyo**
>
> Thank you for carefully reviewing our paper! We greatly appreciate your feedback. Please see below our responses to your comments.
>
> -----
>
> **1. [W.1, R.1] The work is really hard to parse ... rewrite of the sections 2, 3, and 4**
>
> We updated Sections 2, 3, and 4. Please refer to the common response for more details about the changes.
>
> -----
>
> **2. [W.1.1] "BaseTaskSyn" is first mentioned on page 2 but only explained on page 8…**
>
> Thank you for your comment. We moved these figures comparing different techniques to Section 5, where the techniques are explained in detail.
>
> -----
>
> **3. [W.1.3] why you're using imitation learning for part of your method and reinforcement learning for the other part of your method**
>
> For code generation, we use imitation learning, as it has been popularly used in the literature [Bunel et al., 2018]. Token-by-token trajectories for training code prediction are a natural choice.
>
> Conversely, for task generation, the space of possible high-quality traces can be intractably large. As a result, a natural approach is to evaluate the generation process based on the quality of the final task. Our choice is inspired by existing literature where MCTS was used along with a scoring function to generate tasks [Ahmed et al., 2020].
>
> -----
>
> **4. [W1.4] In general, I have no idea what data comes from different systems, how they're trained, and how everything comes together.**
>
> We updated the paper by including additional details in Section 3.3 about the functioning of the puzzle generator. Furthermore, we have incorporated Figure 3 to provide insights into the execution done by the symbolic engine. We updated Figure 4 and the corresponding explanation to depict better the interaction between the RL agent and the environment.
>
> For the code generator, the interaction between its symbolic engine and neural model is somewhat simpler than that of the puzzle generator and more well-studied in literature; hence, we only briefly explain it in Section 3.2 and defer the full design details to Appendix D.1.
>
> The general pipeline of how all the parts are connected, including the input and output of each component, is given in Figure 2.
>
> -----
>
> **5. [W.1.5] There's a mention of a "TaskOracle"...**
>
> TaskOracle takes a code C and generates a task T offline, as mentioned in Section 4.1. It is equivalent to BasePuzzleGen with a large number of rollouts, i.e., $10^6$ (cf. Footnote 2). This process can take a large amount of time, depending on the complexity of C.
>
> BasePuzzleGen is the puzzle-generating technique that uses the unguided base symbolic engine mentioned in Section 4.2. The generation process follows a random search approach, as in [Ghosh et al., 2022].
>
> -----
>
> **6. [W.2, R.2 (b)] Why is it important for anyone to generate these mazes fast?...**
>
> To provide a better motivation for the importance of real-time synthesis, we incorporated Figure 1. We give an overview of an AI system that provides support to a student struggling to solve a task. The AI system helps the student by generating new examples or simple tasks.
>
> We have also included runtime evaluation results in Figure 5c, showing that existing baselines (i.e., BasePuzzleGen and MCTS) are slower and not suitable for real-time synthesis. Moreover, we consider it infeasible to pre-generate tasks, as the space of specifications for which tasks need to be synthesized could be very large, comprising grid patterns, code sketches, and code constraints.
>
> -----
>
> **7. [W.2, R.2 (a)] what is the actual speed difference?**
>
> In Section 4.3 of the revised paper, we conducted a runtime evaluation of our technique with two baselines, i.e., BasePuzzleGen, a symbolic engine based on random search and MCTS puzzle generation introduced in [Ahmed et al., 2020].
>
> Figure 5c shows that NeurPuzzleGen can generate tasks of the required quality with the fewest numbers of rollouts and the fastest runtime. BasePuzzleGen requires more rollouts and thus larger execution times when trying to synthesize tasks of higher quality. MCTS generation has faster runtimes than BasePuzzleGen, but it becomes less scalable for complex tasks of higher quality.
>
> -----
>
> We hope that our responses and the revised paper can address your concerns and are helpful in improving your rating. If you have any other comments or feedback, please let us know! We are looking forward to hearing back from you! Thank you again for the review.
>
> -----
>
> [Ahmed et al., 2020]. "Synthesizing Tasks for Block-based Programming". In NeurIPS, 2020.
>
> [Bunel et al., 2018] "Leveraging Grammar and Reinforcement Learning for Neural Program Synthesis." ICLR 2018.
>
> [Ghosh et al., 2022]. "Adaptive Scaffolding in Block-Based Programming via Synthesizing New Tasks as Pop Quizzes". In AIED, 2022.

---

### Review · Reviewer_Z1kh · 2023-11-13

**Summary Of Contributions:**

The paper proposes an approach for generating visual programming tasks based on a task specification. It first generates a code proposal based on a partial code scaffolding and then generates a visual scene based on code rollouts an a neural decision making model.
The paper demonstrates that this hybrid approach of neural code generation and execution results in higher-quality visual programming tasks than brute-force random search and prompting GPT4 language models.

**Audience:**

Yes

**Claims And Evidence:**

Yes

**Requested Changes:**

# Requested Changes

1) add comparison to prior works on visual programming task synthesis

2) add a proper related work section

3) clarify relation to task oracle and possibly add runtime comparison

**Strengths And Weaknesses:**

# Strengths

The studied task of automatic content generation for education is well-formulated and impactful. The paper is clearly scoped within the domain of visual programming tasks and the resulting tasks are evaluated by human experts, achieving overall high quality.

The paper is clearly written and easy to follow. It evaluates the contributions of the individual components of the approach and compares to relevant baselines, like prompt-engineering GPT4.

It seems the results of the paper could be a useful resource in the programming education community even today.


# Weaknesses

A) No comparison to prior work: the paper cites two relevant prior works on task generation in the tested domains (Ahmed et al., Ghosh et al.), but the proposed approach is not compared to these methods, but instead only to a random search baseline and a GPT4-prompting method.

B) No related work section: the paper lacks a related work section that performs a clear compare & contrast to prior work. The introduction captures some of that, but it would be good to have a separate related work section that summarizes the existing work and may also better justify the (lack of?) comparisons in the experimental section.

C) unclear why we need this if we have an oracle: if I understand correctly, there exists and oracle method for generating tasks (called TASKORACLE in the paper), but it’s downside is that it is slow? It is a bit unclear why this is a major problem since the programming tasks could be generated ahead of time and would only need to be generated once, so even some compute expenses could be tolerated. Given that the paper runs the oracle often to generate the training dataset, it seems feasible to just use the oracle instead of the proposed method? Or is there a detail I am missing? If compute / runtime Is the main benefit of the proposed method over the oracle it would be good to experimentally compare the runtimes.

D) method limited to visual programming domains: the method’s contributions are somewhat limited to the narrow domain of visual programming tasks. The comparisons to GPT4 show that on non-visual domains GPT4 is indeed well-able to generate tasks. Thus, it is likely that the introduced advances would not be necessary in those domains.

---

> ### Author Response · Authors · 2023-11-23
> **Response to Reviewer Z1kh**
>
> Thank you for carefully reviewing our paper! We greatly appreciate your feedback. Please see below our responses to your comments.
>
> -----
>
> **1. No comparison to prior work… on task generation in the tested domains (Ahmed et al., Ghosh et al.).**
>
> In Section 4.3 of the revised paper, we conducted a runtime evaluation of our technique with two baselines. The first baseline is BasePuzzleGen, a symbolic engine based on random search, similar to the technique used in [Ghosh et al., 2022], and the second baseline is MCTS puzzle generation introduced in [Ahmed et al., 2020].
>
> In the evaluation results reported in Figure 5c, we notice that NeurPuzzleGen can generate tasks of the required quality with the fewest numbers of rollouts and the fastest runtime overall. In comparison, we observe that BasePuzzleGen requires more rollouts and thus larger execution times when trying to synthesize tasks of higher quality. Last, MCTS generation has faster runtimes than BasePuzzleGen, but it also becomes less scalable for complex tasks of higher quality.
>
> -----
>
> **2. it would be good to have a separate related work section that summarizes the existing work…**
>
> We thank the reviewer for pointing that out. In Section 1.2 of the revised paper, we added a more comprehensive comparison with existing works on educational task generation. Moreover, we include works related to Large Language Models' spatial and logical reasoning capabilities.
>
> -----
>
> **3. if I understand correctly, there exists an oracle method … but it's downside is that it is slow? It is a bit unclear why this is a major problem since the programming tasks could be generated ahead of time…**
>
> TaskOracle is equivalent to BasePuzzleGen, as mentioned in Footnote 2 of the paper. We have now included a runtime evaluation, depicted in Figure 5c, showing that BasePuzzleGen is slow.
>
> We consider that it is infeasible to generate the tasks ahead of time, as the space of specifications for which tasks need to be synthesized is very large, comprising possible grid patterns, code sketches, and code constraints.
>
> To provide a better motivation for the importance of real-time synthesis, we have incorporated Figure 1 in Section 1. In this figure, we give an overview of an AI system that provides support to a student struggling to solve a given task. The AI system helps the student by generating new examples or simpler tasks.
>
> -----
>
> **4. the method's contributions are somewhat limited to the narrow domain of visual programming tasks. The comparisons to GPT4 show that on non-visual domains GPT4 is indeed well-able to generate tasks. Thus, it is likely that the introduced advances would not be necessary in those domains.**
>
> We would like to point out that GPT-4 struggles with task synthesis for Python programming as well. For instance, [Phung et al., 2023] studied the performance of GPT-4 in six Python programming-related scenarios: program repair, hint generation, grading feedback, pair programming, contextualized explanation, and task synthesis. The performance of GPT-4 for task synthesis is quite low compared to the performance of human tutors.
>
> We note that the main issue here is that the symbolic execution of a code (required for generating test cases or tasks) can be challenging for GPT-4 as has been seen in such contemporary studies (e.g., [Bubeck et al., 2023], [Phung et al., 2023]).
>
> The fact that we are working on visual programming domains further exacerbates these challenges for our setup. For these domains, GPT-4 has to use spatial reasoning, along with programming and logical reasoning. Hence, they can serve as interesting benchmarks for assessing the capabilities of state-of-the-art models.
>
> -----
>
> We hope that our responses and the revised version of the paper can address your concerns and are helpful in improving your rating. If you have any other comments or feedback, please let us know! We are looking forward to hearing back from you! Thank you again for the review.
>
> -----
>
> [Ahmed et al., 2020]. "Synthesizing Tasks for Block-based Programming". In NeurIPS, 2020.
>
> [Bubeck et al., 2023] "Sparks of Artificial General Intelligence: Early experiments with GPT-4". arXiv 2023.
>
> [Ghosh et al., 2022]. "Adaptive Scaffolding in Block-Based Programming via Synthesizing New Tasks as Pop Quizzes". In AIED, 2022.
>
> [Phung et al., 2023] "Generative AI for Programming Education: Benchmarking ChatGPT, GPT-4, and Human Tutors." arXiv 2023.

---

> > ### Comment · Reviewer_Z1kh · 2023-12-21
> > **Thank you**
> >
> > Thanks for answering my review. The response addresses the weaknesses raised in my review and I am happy to recommend acceptance based on validity and relevance as required by the TMLR guidelines.

---

### Author Response · Authors · 2023-11-23
**Response to all reviewers: Revised version of the paper**

Dear Reviewers, Thank you again for your detailed feedback! We have now uploaded a revised version of the paper. We believe that this revision, along with our responses, has addressed most of the reviewers' concerns. We truly believe that the paper has significantly improved based on the feedback, and we really appreciate the reviewers' efforts. We also hope that the revised version is helpful in improving the reviewers' ratings. Below, we provide a summary of the changes:

- **Section 1:** We have incorporated two major updates in this section. First, we have included Figure 1 as an illustration and overview that better highlights the situation where a real-time educational task generation tool is necessary, better motivating our work. Second, we have added Section 1.2 as a related work section, discussing the existing educational content generation techniques and their limitations. This restructuring would provide the reader with a better perspective of the motivation and contributions of our proposed NeurTaskSyn technique.

- **Section 2:** We have updated this section by simplifying the definitions of a task, code, specification, and synthesis objectives. This simplification better captures the generality of our approach.

- **Section 3:** We have updated this section by offering more details about our visual puzzle generation method. We have included Figure 3 to give a better visualization of how the symbolic execution process works. We have also provided more details on the reinforcement learning formulation of the puzzle generation problem in Section 3.2.

- **Section 4:** We have included results for the evaluation w.r.t. runtime metrics in Figure 5c, as requested by the reviewers, comparing our technique to related work (MCTS [Ahmed et al., 2020], BaseTaskSyn [Ghosh et al., 2022]). These results highlight the efficiency of NeurTaskSyn, showing that it is a suitable technique for real-time educational task generation.

Please let us know if you have any other comments or feedback. We will be happy to incorporate further feedback in the final revision. We are looking forward to hearing back from you. Thank you!

[Ahmed et al., 2020]. "Synthesizing Tasks for Block-based Programming". In NeurIPS, 2020.

[Ghosh et al., 2022]. "Adaptive Scaffolding in Block-Based Programming via Synthesizing New Tasks as Pop Quizzes". In AIED, 2022.

---

### Decision · Action_Editor_kxCb · 2023-12-21

**Recommendation:** Accept as is

**Comment:**

The authors have satisfactorily addressed reviewer concerns during the discussion period. The paper may still be challenging to read for much of the TMLR audience, primarily because the application domain is quite different from what ML folks usually consider.

The general consensus from the reviewers was that the work appeared incremental, and that it is unclear whether the paper will have significant impact. However, they all agreed it met the criteria for TMLR acceptance; and, as society-at-large is rapidly imagining AI-assisted futures for education, I find it easy to imagine future work looking to this kind of formal neural task-generation framework for inspiration.

**Audience:**

Yes. All reviewers agree that there is likely someone in the TMLR community for who this is of interest, though all also have highlighted that the work is very niche.

**Claims And Evidence:**

Yes. All reviewers agree the Claims & Evidence criteria are satisfied. The authors have significantly improved the clarity of the presentation during the discussion period.